# General Cutting Planes for Bound-Propagation-Based Neural Network Verification

**Huan Zhang**[*,1]    **Shiqi Wang**[*,2]    **Kaidi Xu**[*,3]

**Linyi Li**[4]    **Bo Li**[4]    **Suman Jana**[2]    **Cho-Jui Hsieh**[5]    **J. Zico Kolter**[1,6]

[1]CMU    [2]Columbia University    [3]Drexel University    [4]UIUC    [5]UCLA    [6]Bosch Center for AI

huan@huan-zhang.com    sw3215@columbia.edu    kx46@drexel.edu
linyi2@illinois.edu    lbo@illinois.edu    suman@cs.columbia.edu
chohsieh@cs.ucla.edu    zkolter@cs.cmu.edu

*\* Equal Contribution*

## Abstract

Bound propagation methods, when combined with branch and bound, are among the most effective methods to formally verify properties of deep neural networks such as correctness, robustness, and safety. However, existing works cannot handle the *general* form of cutting plane constraints widely accepted in traditional solvers, which are crucial for strengthening verifiers with tightened convex relaxations. In this paper, we generalize the bound propagation procedure to allow the addition of arbitrary cutting plane constraints, including those involving relaxed integer variables that do not appear in existing bound propagation formulations. Our generalized bound propagation method, GCP-CROWN, opens up the opportunity to apply *general **c**utting **p**lane methods* for neural network verification while benefiting from the efficiency and GPU acceleration of bound propagation methods. As a case study, we investigate the use of cutting planes generated by off-the-shelf mixed integer programming (MIP) solver. We find that MIP solvers can generate high-quality cutting planes for strengthening bound-propagation-based verifiers using our new formulation. Since the branching-focused bound propagation procedure and the cutting-plane-focused MIP solver can run in parallel utilizing different types of hardware (GPUs and CPUs), their combination can quickly explore a large number of branches with strong cutting planes, leading to strong verification performance. Experiments demonstrate that our method is the first verifier that can *completely* solve the `oval20` benchmark and verify *twice as many* instances on the `oval21` benchmark compared to the best tool in VNN-COMP 2021, and also noticeably outperforms state-of-the-art verifiers on a wide range of benchmarks. GCP-CROWN is part of the $\alpha,\beta$-`CROWN` verifier, **the VNN-COMP 2022 winner**. Code is available at http://PaperCode.cc/GCP-CROWN.

## 1  Introduction

Neural network (NN) verification aims to formally prove or disprove certain properties (e.g., correctness and safety properties) of a NN under a certain set of inputs. These methods can provide worst-case performance guarantees of a NN, and have been applied to mission-critical applications that involve neural networks, such as automatic aircraft control [31, 4], learning-enabled cyber-physical systems [54], and NN based algorithms in an operating system [51].

The NN verification problem is generally NP-complete [30]. For piece-wise linear networks, it can be encoded as a mixed integer programming (MIP) [53] problem with the non-linear ReLU neurons

36th Conference on Neural Information Processing Systems (NeurIPS 2022).

described by binary variables. Thus, fundamentally, the NN verification problem can be solved using the branch and bound (BaB) [10] method similar to generic MIP solvers, by branching some binary variables and relaxing the rest into a convex problem such as linear programming (LP) to obtain bounds on the objective. Although early neural network verifiers relied on off-the-shelf CPU-based LP solvers [36, 9] for bounding in BaB, LP solvers do not scale well to large NNs. Thus, many recent verifiers are instead based on efficient and GPU-accelerated algorithms customized to NN verification, such as bound propagation methods [60, 57], Lagrangian decomposition methods [8, 17] and others [16, 11]. Bound propagation methods, presented in a few different formulations [58, 18, 56, 61, 50, 25], empower state-of-the-art NN verifiers such as $\alpha,\beta$-CROWN [61, 60, 57] and VeriNet [3], and can achieve two to three orders of magnitudes speedup compared to solving the NN verification problem using an off-the-shelf solver directly [57], especially on large networks.

Despite the success of existing NN verifiers, we experimentally find that state-of-the-art NN verifiers may timeout on certain hard instances which a generic MIP solver can solve relatively quickly, sometimes even without branching. Compared to an MIP solver, a crucial factor missing in most scalable NN verifiers is the ability to efficiently generate and solve general cutting planes (or "cuts"). In generic MIP solvers, cutting planes are essential to strengthen the convex relaxation, so that much less branching is required. Advanced cutting planes are among the most important factors in modern MIP solvers [6]; they can strengthen the convex relaxation without removing any valid integer solution from the MIP formulation. In the setting of NN verification, cutting planes reflects complex intra-layer and inter-layer dependencies between multiple neurons, which cannot be easily captured by existing bound propagation methods with single neuron relaxations [45]. This motivates us to seek the combination of efficient bound propagation method with effective cutting planes to further increase the power of NN verifiers.

A few key factors make the inclusion of *general* cutting planes in NN verifiers quite challenging. First, existing efficient bound propagation frameworks such as CROWN [61] and $\beta$-CROWN [57] cannot solve general cutting plane constraints that may involve variables across *different layers* in the MIP formulation. Particularly, these frameworks do not explicitly include the *integer variables* in the MIP formulation that are crucial when encoding many classical strong cutting planes, such as Gomory cuts and mixed integer rounding (MIR) cuts. Furthermore, although some existing works [47, 52, 40] enhanced the basic convex relaxation used in NN verification (such as the Planet relaxation [19]), these enhanced relaxations involve only one or a few neurons in a single layer or two adjacent layers, and are not general enough. In addition, an LP solver is often required to handle these additional cutting plane constraints [40], for which the efficient and GPU-accelerated bound propagation cannot be used, so the use of these tighter relaxations may not always bring improvements.

In this paper, we achieve major progress in using general cutting planes in bound propagation based NN verifiers. To mitigate the challenge of efficiently solving general cuts, **our first contribution** is to generalize existing bound propagation methods to their most general form, enabling constraints involving variables from neurons of any layer as well as integer variables that encode the status of a ReLU neuron. This allows us to consider any cuts during bound propagation without relying on a slow LP solver, and opens up the opportunity for using advanced cutting plane techniques efficiently for the NN verification problem. **Our second contribution** involves combining a cutting-plane-focused, off-the-shelf MIP solver with our GPU-accelerated, branching-focused bound propagation method capable of handling general cuts. We entirely disable branching in the MIP solver and use it only for generating high quality cutting planes not restricting to neurons within adjacent layers. Although an MIP solver often cannot verify large neural networks, we find that they can generate high quality cutting planes within a short time, significantly helping bound propagation to achieve better bounds.

Our experiments show that general cutting planes can bring significant improvements to NN verifiers: we are the first verifier that *completely* solves *all* instances in the oval20 benchmark, with an average time of *less than 5 seconds* per instance; on the even harder oval21 benchmark in VNN-COMP 2021 [3], we can verify *twice as many instances* compared to the competition winner. We also outperform existing state-of-the-art bound-propagation-based methods including those using multi-neuron relaxations [20] (a limited form of cutting planes).

## 2   Background

**The NN verification problem**   We consider the verification problem for an $L$-layer ReLU-based Neural Network (NN) with inputs $\hat{x}^{(0)} := x \in \mathbb{R}^{d_0}$, weights $\mathbf{W}^{(i)} \in \mathbb{R}^{d_i \times d_{i-1}}$, and biases

$\mathbf{b}^{(i)} \in \mathbb{R}^{d_i}$ ($i \in \{1, \cdots, L\}$). We can get the NN outputs $f(\boldsymbol{x}) = \boldsymbol{x}^{(L)} \in \mathbb{R}^{d_L}$ by sequentially propagating the input $\boldsymbol{x}$ through affine layers with $\boldsymbol{x}^{(i)} = \mathbf{W}^{(i)}\hat{\boldsymbol{x}}^{(i-1)} + \mathbf{b}^{(i)}$ and ReLU layer with $\hat{\boldsymbol{x}}^{(i)} = \text{ReLU}(\boldsymbol{x}^{(i)})$. We also let scalars $\hat{x}_j^{(i)}$ and $x_j^{(i)}$ denote the post-activation and pre-activation, respectively, of $j$-th ReLU neuron in $i$-th layer. Throughout the paper, we use bold symbols to denote vectors (e.g., $\boldsymbol{x}^{(i)}$) and regular symbols to denote scalars (e.g., $x_j^{(i)}$ is the $j$-th element of $\boldsymbol{x}^{(i)}$). We use the shorthand $[N]$ to denote $\{1, \cdots, N\}$, and $\mathbf{W}_{:,j}^{(i)}$ is the $j$-th column of $\mathbf{W}^{(i)}$.

Commonly, the input $\boldsymbol{x}$ is bounded within a perturbation set $\mathcal{C}$ (such as an $\ell_p$ norm ball) and the verification specification defines a property of the output $f(\boldsymbol{x})$ that should hold for any $\boldsymbol{x} \in \mathcal{C}$, e.g., whether the true label's logit $f_y(\boldsymbol{x})$ will be always larger than another label's logit $f_j(\boldsymbol{x})$, (i.e., checking if $f_y(\boldsymbol{x}) - f_j(\boldsymbol{x})$ is always positive). Since we can append the verification specification (such as a linear function on neural network output) as an additional layer of the network, canonically, the NN verification problem requires one to solve the following one-dimensional ($d_L = 1$) optimization objective on $f(\boldsymbol{x})$:

$$f^* = \min_{\boldsymbol{x}} f(\boldsymbol{x}), \forall \boldsymbol{x} \in \mathcal{C} \tag{1}$$

with the relevant property defined to be proven if the optimal solution $f^* \geq 0$. Throughout this work, we consider the $\ell_\infty$ norm ball $\mathcal{C} := \{\boldsymbol{x} : \|\boldsymbol{x} - \boldsymbol{x}_0\|_\infty \leq \epsilon\}$ where $\boldsymbol{x}_0$ is a predefined constant (e.g., a clean input image), although it is possible to extend to other norms or specifications [43, 59].

**The MIP and LP formulation for NN verification**  The mixed integer programming (MIP) formulation is the root of many NN verification algorithms. This formulation uses binary variables $\mathbf{z}$ to encode the non-linear ReLU neurons to make the non-convex optimization problem (1) tractable. Additionally, we assume that we know sound pre-activation bounds $\boldsymbol{l}^{(i)} \leq \boldsymbol{x}^{(i)} \leq \boldsymbol{u}^{(i)}$ for $\boldsymbol{x} \in \mathcal{C}$ which can be obtained via cheap bound propagation methods such as IBP [23] or CROWN [61]. Then ReLU neurons for each layer $i$ can be classified into three classes [58], namely "active" ($\mathcal{I}^{+(i)}$), "inactive" ($\mathcal{I}^{-(i)}$) and "unstable" ($\mathcal{I}^{(i)}$) neurons, respectively:

$$\mathcal{I}^{+(i)} := \{j : l_j^{(i)} \geq 0\}; \quad \mathcal{I}^{-(i)} := \{j : u_j^{(i)} \leq 0\}; \quad \mathcal{I}^{(i)} := \{j : l_j^{(i)} \leq 0, \ u_j^{(i)} \geq 0\}$$

Based on the definition of ReLU, activate and inactive neurons are linear functions, so only unstable neurons require binary encoding. The *MIP formulation* of (1) is:

$$f^* = \min_{\boldsymbol{x}, \hat{\boldsymbol{x}}, \mathbf{z}} f(\boldsymbol{x}) \qquad \text{s.t. } f(\boldsymbol{x}) = \boldsymbol{x}^{(L)}; \quad \hat{\boldsymbol{x}}^{(0)} = \boldsymbol{x}; \quad \boldsymbol{x} \in \mathcal{C}; \tag{2}$$

$$\boldsymbol{x}^{(i)} = \mathbf{W}^{(i)}\hat{\boldsymbol{x}}^{(i-1)} + \mathbf{b}^{(i)}; \quad i \in [L], \tag{3}$$

$$\hat{x}_j^{(i)} \geq 0; \quad j \in \mathcal{I}^{(i)}, i \in [L-1] \tag{4}$$

$$\hat{x}_j^{(i)} \geq x_j^{(i)}; \quad j \in \mathcal{I}^{(i)}, i \in [L-1] \tag{5}$$

$$\hat{x}_j^{(i)} \leq u_j^{(i)} z_j^{(i)}; \quad j \in \mathcal{I}^{(i)}, i \in [L-1] \tag{6}$$

$$\hat{x}_j^{(i)} \leq x_j^{(i)} - l_j^{(i)}(1 - z_j^{(i)}); \quad j \in \mathcal{I}^{(i)}, i \in [L-1] \tag{7}$$

$$z_j^{(i)} \in \{0, 1\}; \quad j \in \mathcal{I}^{(i)}, i \in [L-1] \tag{8}$$

$$\hat{x}_j^{(i)} = x_j^{(i)}; \quad j \in \mathcal{I}^{+(i)}, i \in [L-1] \tag{9}$$

$$\hat{x}_j^{(i)} = 0; \quad j \in \mathcal{I}^{-(i)}, i \in [L-1] \tag{10}$$

Since a MIP problem is slow or intractable to solve, it is commonly relaxed as a When the integer variables are relaxed to continuous ones, we obtain the *LP relaxation* of NN verification problem:

$$f_{\text{LP}}^* = \min_{\boldsymbol{x}, \hat{\boldsymbol{x}}, \mathbf{z}} f(\boldsymbol{x})$$

$$\text{s.t. } (3), (2), (4), (5), (6), (7), (9), (10), \quad 0 \leq z_j^{(i)} \leq 1; \quad j \in \mathcal{I}^{(i)}, \ i \in [L-1] \tag{11}$$

The ReLU constraints involving $z$ is often projected out, leading to the well-known *Planet relaxation* used in many NN verifiers, replacing (6), (7) and (8) with a single constraint to get an equivalent LP:

$$f_{\text{LP}}^* = \min_{\boldsymbol{x}, \hat{\boldsymbol{x}}, \mathbf{z}} f(\boldsymbol{x})$$

$$\text{s.t. } (3), (2), (4), (5), (9), (10), \quad \hat{x}_j^{(i)} \leq \frac{u_j^{(i)}}{u_j^{(i)} - l_j^{(i)}}(x_j^{(i)} - l_j^{(i)}); \ j \in \mathcal{I}^{(i)}, \ i \in [L-1] \tag{12}$$

Due to the relaxations, the objective of the LP formulation is always a lower bound of the MIP formulation: $f_{LP}^* \leq f^*$. A verifier using this formulation is incomplete: if $f_{LP}^* \geq 0$, then $f^* \geq 0$ and the property is verified; otherwise, we cannot conclude the sign of $f^*$ so the verifier must return "unknown". Branch and bound can be used to improve the lower bound and achieve completeness [10, 57]. However, in this paper, we work on an orthogonal direction of strengthening the LP formulation by adding cutting planes to obtain larger bounds.

**Bound propagation methods** Instead of solving the LP formulation directly using a LP solver, bound propagation methods aims to quickly give a lower bound for $f_{LP}^*$. For example, CROWN [61] and $\beta$-CROWN [57] propagate a sound linear lower bound backwards for $f_L(\boldsymbol{x})$ with respect to each intermediate layer. For example, suppose we know

$$\min_{\boldsymbol{x} \in \mathcal{C}} f_L(\boldsymbol{x}) \geq \min_{\boldsymbol{x} \in \mathcal{C}} \mathbf{a}^{(i)^\top} \hat{\boldsymbol{x}}^{(i)} + c^{(i)} \tag{13}$$

With $i = L - 1$ the above is trivially hold with $\mathbf{a}^{(L-1)} = \mathbf{W}^{(L)}$, $c^{(L-1)} = \mathbf{b}^{(L)}$. In bound propagation methods, a propagation rule propagates an inequality (13) through a previous layer $\hat{\boldsymbol{x}}^{(i)} := \text{ReLU}(\mathbf{W}^{(i)}\hat{\boldsymbol{x}}^{(i-1)} + \mathbf{b}^{(i)})$ to obtain a *sound* inequality with respect to $\hat{\boldsymbol{x}}^{(i-1)}$:

$$\min_{\boldsymbol{x} \in \mathcal{C}} f_L(\boldsymbol{x}) \geq \min_{\boldsymbol{x} \in \mathcal{C}} \mathbf{a}^{(i-1)^\top} \hat{\boldsymbol{x}}^{(i-1)} + c^{(i-1)}$$

Here $\mathbf{a}^{(i-1)}$, $c^{(i-1)}$ can be calculated in close-form via $\mathbf{a}^{(i)}$, $c^{(i)}$, $\mathbf{W}^{(i)}$, $\mathbf{b}^{(i)}$, $\boldsymbol{l}^{(i)}$ and $\boldsymbol{u}^{(i)}$ such that the bound still holds (see Lemma 1 in [57]). Applying the procedure repeatedly will eventually reach the input layer:

$$\min_{\boldsymbol{x} \in \mathcal{C}} f_L(\boldsymbol{x}) \geq \min_{\boldsymbol{x} \in \mathcal{C}} \mathbf{a}^{(0)^\top} \boldsymbol{x} + c^{(0)} \tag{14}$$

The minimization on linear layer can be solved easily when $\mathcal{C}$ is a $\ell_p$ norm ball to obtain a valid lower bound of $f^*$. Since the bounds propagate layer-by-layer, this process can be implemented efficiently on GPUs [59] without relying on a slow LP solver, which greatly improves the scalability and solving time. Additionally, it is often used to obtain intermediate layer bounds $\boldsymbol{l}^{(i)}$ and and $\boldsymbol{u}^{(i)}$ required for the MIP formulation (6)(7), by treating each $x_j^{(i)}$ as the output neuron. The bound propagation rule can either be derived in primal space [61], dual space [58] or abstract interpretations [50]. In Sec.3.1, we will discuss the our bound propagation procedure with general cutting plane constraints.

## 3 Neural Network Verification with General Cutting Planes

### 3.1 GCP-CROWN: General Cutting Planes in Bound Propagation

In this section, we generalize existing bound propagation method to handle general cutting plane constraints. Our goal is to derive a bound propagation rule similar to CROWN and $\beta$-CROWN discussed in Section 2, however considering additional constraints among any variables within the LP relaxation. To achieve this, we first derive the dual problem of the LP; inspired by the dual formulation, we derive the bound prorogation rule in a layer by layer manner that takes all cutting plane constraints into consideration. The derivation process is inspired by [58, 45] and [57].

**LP relaxation with cutting planes.** In this section, we derive the bound propagation procedure under the presence of general cutting plane constraints. A cutting plane is a constraint involving any variables $\boldsymbol{x}^{(i)}$ (pre-activation), $\hat{\boldsymbol{x}}^{(i)}$ (post-activation), $\mathbf{z}^{(i)}$ (ReLU indicators) from any layer $i$:

$$\sum_{i=1}^{L-1} \left( \boldsymbol{h}^{(i)^\top} \boldsymbol{x}^{(i)} + \boldsymbol{g}^{(i)^\top} \hat{\boldsymbol{x}}^{(i)} + \boldsymbol{q}^{(i)^\top} \mathbf{z}^{(i)} \right) \leq d$$

Here $\boldsymbol{h}^{(i)}$, $\boldsymbol{g}^{(i)}$ and $\boldsymbol{q}^{(i)}$ are coefficients for this cut constraint. The difference between a valid cutting plane and an arbitrary constraint is that a valid cutting plane should not remove any valid integer solution from the MIP formulation. Our new bound propagation procedure can work for any constraints, although in this work we focus on studying the impacts of cutting planes. When there are $N$ cutting planes, we write them in a matrix form:

$$\sum_{i=1}^{L-1} \left( \boldsymbol{H}^{(i)} \boldsymbol{x}^{(i)} + \boldsymbol{G}^{(i)} \hat{\boldsymbol{x}}^{(i)} + \boldsymbol{Q}^{(i)} \mathbf{z}^{(i)} \right) \leq \boldsymbol{d} \tag{15}$$

where $\boldsymbol{H}^{(i)}, \boldsymbol{G}^{(i)}, \boldsymbol{Q}^{(i)} \in \mathbb{R}^{N \times d_i}$. The LP relaxation with all cutting planes is:

$$f_{\text{LP-cut}}^* = \min_{\boldsymbol{x}, \hat{\boldsymbol{x}}, \mathbf{z}} f(x)$$

$$\text{s.t. } (3), (2), (4), (5), (6), (7), (9), (10), \quad 0 \le z_j^{(i)} \le 1; \quad j \in \mathcal{I}^{(i)}, \ i \in [L-1]$$

$$\sum_{i=1}^{L-1} \left( \boldsymbol{H}^{(i)} \boldsymbol{x}^{(i)} + \boldsymbol{G}^{(i)} \hat{\boldsymbol{x}}^{(i)} + \boldsymbol{Q}^{(i)} \mathbf{z}^{(i)} \right) \le \boldsymbol{d} \tag{16}$$

Since an additional constraint is added, $f_{\text{LP-cut}}^* \ge f_{\text{LP}}^*$ and we get closer to $f^*$. Unlike the original LP where each constraint only contains variables from two consecutive layers, our general cutting plane constraint may involve any variable from any layer in a single constraint.

**The dual problem with cutting planes** We first show the dual problem for the above LP. The dual problem we consider here is different from existing works in two ways: first, we have constraints with integer variables to support potential cutting planes on $\mathbf{z}$. Additionally, we have cutting planes constraints that may involve variables in *any* layer, so the dual in previous works such as [58] cannot be directly reused. Our dual problem is given below (derivation details in Appendix A):

$$f_{\text{LP-cut}}^* = \max_{\substack{\boldsymbol{\nu}, \boldsymbol{\mu} \ge 0, \boldsymbol{\tau} \ge 0 \\ \boldsymbol{\gamma} \ge 0, \boldsymbol{\pi} \ge 0, \boldsymbol{\beta} \ge 0}} -\epsilon \| \boldsymbol{\nu}^{(1)\top} \mathbf{W}^{(1)} \boldsymbol{x}_0 \|_1 - \boldsymbol{\beta}^\top \boldsymbol{d} - \sum_{i=1}^{L} \boldsymbol{\nu}^{(i)\top} \mathbf{b}^{(i)}$$

$$+ \sum_{i=1}^{L-1} \sum_{j \in \mathcal{I}^{(i)}} \left[ \pi_j^{(i)} l_j^{(i)} - \text{ReLU}(u_j^{(i)} \gamma_j^{(i)} + l_j^{(i)} \pi_j^{(i)} - \boldsymbol{\beta}^\top \boldsymbol{Q}_{:,j}^{(i)}) \right]$$

s.t. $\boldsymbol{\nu}^{(L)} = -1$; and for each $i \in [L-1]$ :

$$\nu_j^{(i)} = \boldsymbol{\nu}^{(i+1)\top} \mathbf{W}_{:,j}^{(i+1)} - \boldsymbol{\beta}^\top (\boldsymbol{H}_{:,j}^{(i)} + \boldsymbol{G}_{:,j}^{(i)}); \ j \in \mathcal{I}^{+(i)}$$

$$\nu_j^{(i)} = -\boldsymbol{\beta}^\top \boldsymbol{H}_{:,j}^{(i)}; \ j \in \mathcal{I}^{-(i)}$$

and for each $j \in \mathcal{I}^{(i)}$ the two equalities below hold:

$$\nu_j^{(i)} = \pi_j^{(i)} - \tau_j^{(i)} - \boldsymbol{\beta}^\top \boldsymbol{H}_{:,j}^{(i)}; \quad \left(\pi_j^{(i)} + \gamma_j^{(i)}\right) - \left(\mu_j^{(i)} + \tau_j^{(i)}\right) = \boldsymbol{\nu}^{(i+1)\top} \mathbf{W}_{:,j}^{(i+1)} - \boldsymbol{\beta}^\top \boldsymbol{G}_{:,j}^{(i)}$$

Instead of solving the dual problem exactly, we use it to obtain a lower bound of $f_{\text{LP-cut}}^*$. Intuitively, due to the definition of due problem, any valid setting of dual variables leads to a lower bound of $f_{\text{LP-cut}}^*$. Informally, starting from $\boldsymbol{\nu}^{(L)} = -1$, by applying the constraints in this dual formulation, we can compute $\boldsymbol{\nu}^{(L-1)}, \boldsymbol{\nu}^{(L-2)}, \cdots$ until $\boldsymbol{\nu}^{(1)}$. The final objective is a function of $\boldsymbol{\nu}^{(i)}$, $i \in [N]$ and other dual variables. Precisely, our GCP-CROWN bound propagation procedure with general cutting plane constraint is presented in the theorem below (proof in Appendix A):

**Theorem 3.1** (Bound propagation with general cutting planes)**.** *Given any optimizable parameters* $0 \le \alpha_j^{(i)} \le 1$ *and* $\boldsymbol{\beta} \ge 0$, $f_{\text{LP-cut}}^*$ *is lower bounded by the following objective function,* $\pi_j^{(i)*}$ *is a function of* $\boldsymbol{Q}_{:,j}^{(i)}$:

$$g(\boldsymbol{\alpha}, \boldsymbol{\beta}) = -\epsilon \| \boldsymbol{\nu}^{(1)\top} \mathbf{W}^{(1)} \boldsymbol{x}_0 \|_1 - \sum_{i=1}^{L} \boldsymbol{\nu}^{(i)\top} \mathbf{b}^{(i)} - \boldsymbol{\beta}^\top \boldsymbol{d} + \sum_{i=1}^{L-1} \sum_{j \in \mathcal{I}^{(i)}} h_j^{(i)}(\boldsymbol{\beta})$$

*where variables* $\boldsymbol{\nu}^{(i)}$ *are obtained by propagating* $\boldsymbol{\nu}^{(L)} = -1$ *throughout all* $i \in [L-1]$:

$$\nu_j^{(i)} = \boldsymbol{\nu}^{(i+1)\top} \mathbf{W}_{:,j}^{(i+1)} - \boldsymbol{\beta}^\top (\boldsymbol{H}_{:,j}^{(i)} + \boldsymbol{G}_{:,j}^{(i)}), \ j \in \mathcal{I}^{+(i)}$$

$$\nu_j^{(i)} = -\boldsymbol{\beta}^\top \boldsymbol{H}_{:,j}^{(i)}, \ j \in \mathcal{I}^{-(i)}$$

$$\nu_j^{(i)} = \pi_j^{(i)*} - \alpha_j^{(i)} [\hat{\nu}_j^{(i)}]_- - \boldsymbol{\beta}^\top \boldsymbol{H}_{:,j}^{(i)}, \ j \in \mathcal{I}^{(i)}$$

*Here* $\hat{\nu}_j^{(i)}$, $\pi_j^{(i)*}$ *and* $h_j^{(i)}(\boldsymbol{\beta})$ *are defined for each unstable neuron* $j \in \mathcal{I}^{(i)}$.

$$\hat{\nu}_j^{(i)} := \boldsymbol{\nu}^{(i+1)\top} \mathbf{W}_{:,j}^{(i+1)} - \boldsymbol{\beta}^\top \boldsymbol{G}_{:,j}^{(i)}$$

$$\pi_j^{(i)*} = \max \left( \min \left( \frac{u_j^{(i)} [\hat{\nu}_j^{(i)}]_+ + \boldsymbol{\beta}^\top \boldsymbol{Q}_{:,j}^{(i)}}{u_j^{(i)} - l_j^{(i)}}, [\hat{\nu}_j^{(i)}]_+ \right), 0 \right)$$

$$h_j^{(i)}(\boldsymbol{\beta}) = \begin{cases} l_j^{(i)}\pi_j^{(i)^*} & \text{if } l_j^{(i)}[\hat{\nu}_j^{(i)}]_+ \leq \boldsymbol{\beta}^\top \boldsymbol{Q}_{:,j}^{(i)} \leq u_j^{(i)}[\hat{\nu}_j^{(i)}]_+ \\ 0 & \text{if } \boldsymbol{\beta}^\top \boldsymbol{Q}_{:,j}^{(i)} \geq u_j^{(i)}[\hat{\nu}_j^{(i)}]_+ \\ \boldsymbol{\beta}^\top \boldsymbol{Q}_{:,j}^{(i)} & \text{if } \boldsymbol{\beta}^\top \boldsymbol{Q}_{:,j}^{(i)} \leq l_j^{(i)}[\hat{\nu}_j^{(i)}]_+ \end{cases}$$

$$\pi_j^{(i)^*} \ is \ a \ function \ of \ \boldsymbol{Q}_{:,j}^{(i)}$$

Based on Theorem 3.1, to obtain an lower bound of $f_{\text{LP-cut}}^*$, we start with any valid setting of $0 \leq \boldsymbol{\alpha} \leq 1$ and $\boldsymbol{\beta} \geq 0$ and $\boldsymbol{\nu}^{(L)} = -1$. According to the bound propagation rule, we can compute each $\boldsymbol{\nu}^{(i)}, i \in [L-1]$, in a layer by layer manner. Then objective $g(\boldsymbol{\alpha}, \boldsymbol{\beta})$ can be evaluated based on all $\boldsymbol{\nu}^{(i)}$ to give an lower bound of $f_{\text{LP-cut}}^*$. Since any valid setting of $\boldsymbol{\alpha}$ and $\boldsymbol{\beta}$ lead to a valid lower bound, we can optimize $\boldsymbol{\alpha}$ and $\boldsymbol{\beta}$ using gradient ascent in a similar manner as in [60, 57] to tighten this lower bound. The entire procedure can also run on GPU for great acceleration.

**Connection to Convex Outer Adversarial Polytope**   In convex outer adversarial polytope [58], a bound propagation rule was developed in a similar manner in the dual space without considering cutting plane constraints, and is a special case of ours. We denote their bound propagation objective function as $g_{\text{WK}}$ which also contains optimizable parameters $\boldsymbol{\alpha}_{\text{WK}}$.

**Proposition 3.2.** *Given the same input $\boldsymbol{x}$, perturbation set $\mathcal{C}$, network weights, and $N$ cutting plane constraints,*

$$\max_{\boldsymbol{\alpha}, \boldsymbol{\beta}} g(\boldsymbol{\alpha}, \boldsymbol{\beta}) \geq \max_{\boldsymbol{\alpha}_{\text{WK}}} g_{\text{WK}}(\boldsymbol{\alpha}_{\text{WK}})$$

*Proof.* In Theorem 3.1, when all $\boldsymbol{\beta}$ are set to 0, then $\pi_j^{(i)^*} = \frac{u_j^{(i)}[\hat{\nu}_j^{(i)}]_+}{u_j^{(i)} - l_j^{(i)}}$ and $h_j^{(i)}(\boldsymbol{\beta}) = \pi_j^{(i)^*} l_j^{(i)}$, we recover exactly the same bound propagation equations as in [58]. However, since we allow the addition of cutting plane methods and we can maximize over the additional parameter $\boldsymbol{\beta}$, the objective given by our bound propagation is always at least as good as $g_{\text{WK}}$. $\qquad\square$

**Connection to CROWN-like bound propagation methods**   CROWN [61] and $\alpha$-CROWN [60] use the same bound propagation rule as [58] so Proposition 3.2 also applies, although they were derived from primal space without explicitly formulating the problem as a LP. Salman et al. [45] showed that many other bound propagation methods [50, 55] are equivalent to or weaker than [58]. Recently, Wang et al. [57] extends CROWN to $\beta$-CROWN to handle split constraints (e.g., $x_j^{(i)} \geq 0$). It can be seen a special case as GCP-CROWN where all $\boldsymbol{H}$, $\boldsymbol{G}$ and $\boldsymbol{Q}$ matrices are zeros except:

$$\boldsymbol{H}_{j,j}^{(i)} = 1, j \in \mathcal{I}^{-(i)} \qquad \text{for } x_j^{(i)} \leq 0 \text{ split}; \qquad \boldsymbol{H}_{j,j}^{(i)} = -1, j \in \mathcal{I}^{+(i)} \qquad \text{for } x_j^{(i)} \geq 0 \text{ split}$$

In addition, [20] encodes multi-neuron relaxations using sparse $\boldsymbol{H}^{(i)}$ and $\boldsymbol{G}^{(i)}$ and each cut contains a small number of neurons involving $\boldsymbol{x}^{(i)}$ and $\hat{\boldsymbol{x}}^{(i)}$ for the same layer $i$. Wang et al. [57] derived bound propagation rules from both the dual LP and the primal space with a Lagrangian without LP. However, in our case, it is not intuitive to derive bound propagation without LP due to the potential cutting planes on relaxed integer variables **z**, which do not appear without the explicit LP formulation. Furthermore, although we derived cutting planes for bound propagation methods, it is technically also possible to derive them using other bounding frameworks such as Lagrangian decomposition [8].

### 3.2   Branch-and-bound with GCP-CROWN and MIP Solver Generated Cuts

To build a complete NN verifier, we follow the popular branch-and-bound (BaB) procedure [10, 9] in state-of-the-art NN verifiers with GPU accelerated bound propagation method [8, 60, 16, 57], and our GCP-CROWN is used as the bounding procedure in BaB. We refer the readers to Appendix B for a more detailed background on branch-and-bound. Having the efficient bound propagation procedure with general cutting plane constraints, we now need to find a good set of general cutting planes $\boldsymbol{H}^{(i)}, \boldsymbol{G}^{(i)}, \boldsymbol{Q}^{(i)}$ to accelerate NN verification. Since GCP-CROWN can adopt any cutting planes, to fully exploit its power, we propose to use off-the-shelf MIP solvers to generate cutting planes and create an NN verifier combining GPU-accelerated bound propagation with strong cuts generated by a MIP solver. We make the bound propagation on GPU and the MIP solver on CPU run in parallel with cuts added on-the-fly, so the original strong performance of bound-propagation-based

NN verifier will never be affected by a potentially slow MIP solver. This allows us to make full use of available computing resource (GPU + CPU). The architecture of our verifier is shown in Fig 1.

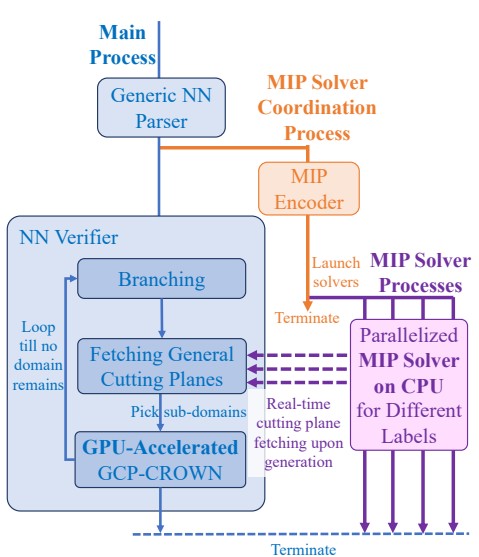

Figure 1: Overview of our cutting-plane-enhanced, fully-parallelized NN verifier.

**MIP solvers for cutting plane generation** Generic MIP solvers such as cplex[28] and gurobi[24] also apply a branch-and-bound strategy, conceptually similar to state-of-the-art NN verifiers. They often tend to be slower than specialized NN verifiers because MIP solvers rely on slower bounding procedures (e.g., Simplex or barrier method) and cannot apply an GPU-accelerated method such as bound propagation or Lagrangian decomposition. However, we still find that MIP solvers are a strong baseline when *combined with tight intermediate layer bounds*. For example, in the oval21 benchmark in Table 2, $\alpha$-CROWN+MIP (MIP solver combined with tight intermediate layer bound computed by $\alpha$-CROWN) is able to solve 4 more instances compared to all other tools in the competition. Our investigation found that $\alpha$-CROWN+MIP explores much less branches than other state-of-the-art branch-and-bound based NN verifiers, however before branching starts, the MIP solver produces very effective cutting planes that can often verify an instance with little branching. MIP solvers is able to discover sophisticated cutting planes involving several NN layers reflecting complex correlations among neurons, while existing NN verifiers only exploit specific forms of constraints within same layer or between adjacent layers [52, 47, 40]. This motivates us to combine the strong cutting planes generated by MIP solvers with GPU-accelerated bound-propagation-based BaB.

In this work, we use cplex as the MIP solver for cutting plane generation since gurobi cannot export cuts. We entirely disable all branching features in cplex and make it focus on finding cutting planes only. These cutting planes are generated only for the root node of the BaB search tree so they are sound for any subdomains with neuron splits in BaB. We conduct branching using our generalized bound propagation procedure in Section (3.1) with the cutting planes generated by cplex.

**Fully-parallelized NN verifier design** We design our verifier as shown in Figure 1. After parsing the NN under verification, we launch a separate process to encode the NN verification problem as MIP problem and start multiple MIP solvers, one for each target label to be verified. At the same time, the main NN verifier process executes branch-and-bound without waiting for the MIP solver process. In each iteration of branch and bound, we query the MIP solving processes and fetch any newly generated cutting planes. If any cutting planes are produced, they are added as $\boldsymbol{H}^{(i)}, \boldsymbol{G}^{(i)}, \boldsymbol{Q}^{(i)}$ in GCP-CROWN and tighten the bounds for subsequent branching and bounding. If no cutting planes are produced, GCP-CROWN reduces to $\beta$-CROWN [57]. Since our verifier is based on strengthening the bounds in $\beta$-CROWN with sound cutting planes, it is also sound and complete.

**Adjustments to existing branch and bound method.** We implement GCP-CROWN into the $\alpha,\beta$-CROWN verifier [61, 59, 57], the winning verifier in VNN-COMP 2021 [3], as the backbone for BaB with bound propagation. To better exploit the power of cutting planes under a fully parallel and asynchronous design, we made a key changes to the BaB procedure. When the number of BaB subdomains are greater than batch size, we rank the subdomains by their lower bounds and choose the *easiest domains* with largest lower bounds first to verify with GCP-CROWN, unlike most existing verifiers which solve the worst domains first. We use such a order because the MIP solver generates cutting planes incrementally. Solving these easier subdomains tend to require no or fewer cutting planes, so we solve them at earlier stages where cutting planes have not been generated or are relatively weak. On the other hand, if we split worst subdomains first, the number of subdomains will grow quickly, and it can take a long time to verify these domains when stronger cuts become available later. Under a similar rationale, when verifying a multi-class model and BaB needs to verify each

Table 1: Average runtime and average number of branches on `oval20` benchmarks with 100 properties per model. Timeout is set to 3,600 seconds (consistent with other literature results). GCP-CROWN is the only method that can completely solve *all* instances (0% timeout) and the average time per-instance is less than 5 seconds on all three networks.

| | CIFAR-10 Base | | | CIFAR-10 Wide | | | CIFAR-10 Deep | | |
|---|---|---|---|---|---|---|---|---|---|
| Method | time(s) | branches | %timeout | time(s) | branches | %timeout | time(s) | branches | %timeout |
| MIPplanet [19] | 2849.69 | - | 68.00 | 2417.53 | - | 46.00 | 2302.25 | - | 40.00 |
| BaBSR [9] | 2367.78 | 1020.55 | 36.00 | 2871.14 | 812.65 | 49.00 | 2750.75 | 401.28 | 39.00 |
| GNN-online [36] | 1794.85 | 565.13 | 33.00 | 1367.38 | 372.74 | 15.00 | 1055.33 | 131.85 | 4.00 |
| BDD+ BaBSR [8] | 807.91 | 195480.14 | 20.00 | 505.65 | 74203.11 | 10.00 | 266.28 | 12722.74 | 4.00 |
| Fast-and-Complete [60] | 695.01 | 119522.65 | 17.00 | 495.88 | 80519.85 | 9.00 | 105.64 | 2455.11 | 1.00 |
| OVAL (BDD+ GNN)*[8, 36] | 662.17 | 67938.38 | 16.00 | 280.38 | 17895.94 | 6.00 | 94.69 | 1990.34 | 1.00 |
| A.set BaBSR [16] | 381.78 | 12004.60 | 7.00 | 165.91 | 2233.10 | 3.00 | 190.28 | 2491.55 | 2.00 |
| BigM+A.set BaBSR [16] | 390.44 | 11938.75 | 7.00 | 172.65 | 4050.59 | 3.00 | 177.22 | 3275.25 | 2.00 |
| BaDNB (BDD+ FSB)[17] | 309.29 | 38239.04 | 7.00 | 165.53 | 11214.44 | 4.00 | 10.50 | 368.16 | 0.00 |
| ERAN*[47, 48, 50, 49] | 805.94 | - | 5.00 | 632.20 | - | 9.00 | 545.72 | - | 0.00 |
| $\beta$-CROWN [57] | 118.23 | 208018.21 | 3.00 | 78.32 | 116912.57 | 2.00 | 5.69 | 41.12 | 0.00 |
| $\alpha$-CROWN+MIP† | 335.50 | 8523.37 | 3.00 | 203.87 | 2029.60 | **0.00** | 76.90 | 1364.24 | **0.00** |
| GCP-CROWN with MIP cuts | **4.07** | 2580.53 | **0.00** | **3.02** | 2095.18 | **0.00** | **3.87** | 110.92 | **0.00** |

* Results from VNN-COMP 2020 report [34].   † A new baseline proposed and evaluated in this work, not presented in previous papers.

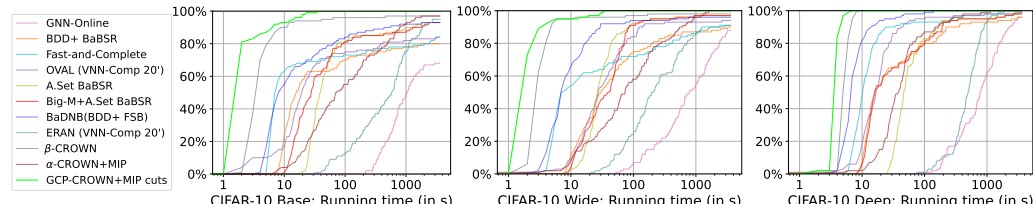

Figure 2: Percentage of solved properties on the `oval20` benchmark vs. running time (timeout 1 hour).

target class label one by one, we start BaB from the easiest label first ($\alpha$-CROWN bound closest to 0), allowing the MIP solver to run longer for harder labels, generating stronger cuts for harder labels.

## 4   Experiments

We now evaluate our verifier, GCP-CROWN with MIP cuts, on a few popular verification benchmarks. Since our verifier uses a MIP solver in our pipeline, we also include a new baseline, $\alpha$-CROWN+MIP, which uses `gurobi` as the MIP solver with the tightest possible intermediate layer bounds from $\alpha$-CROWN [59]. We use the same branch and bound algorithm as in $\beta$-CROWN and we use filtered smart branching (FSB) [16] as the branching heuristic in all experiments. Without cutting planes, GCP-CROWN becomes vanilla $\beta$-CROWN as we share the same code base as $\beta$-CROWN. We include model information and detailed setup for our experiments in Appendix C. GCP-CROWN has been integrated into the $\alpha,\beta$-`CROWN` (alpha-beta-CROWN) verifier, and the instructions to reproduce results in this paper are available at http://PaperCode.cc/GCP-CROWN.

**Results on the `oval20` benchmark in VNN-COMP 2020.**   `oval20` is a popular benchmark consistently used in huge amount of NN verifiers and it perfectly reflects the progress of NN verifiers. We include literature results for many baselines in Table 1. We are the only verifier that can *completely solve* all three models without any timeout. Our average runtime is significantly lower compared to the time of baselines because we have no timeout (counted as 3600s), and our slowest instance only takes about a few minutes while easy ones only take a few seconds, as shown in Figure 2. Additionally, we often use less number of branches compared to the state-of-the-art verifier, $\beta$-CROWN, since our strong cutting planes help us to eliminate many hard to solve subdomains in BaB. Furthermore, we highlight that $\alpha$-CROWN+MIP also achieves a low timeout rate, although it is much slower than our bound propagation based approach combined with cuts from a MIP solver.

**Results on VNN-COMP 2021 benchmarks.**   Among the eight scoring benchmarks in VNN-COMP 2021 [3], only two (`oval21` and `cifar10-resnet`) are most suitable for the evaluation of this work. Among other benchmarks, `acasxu` and `nn4sys` have low input dimensionality and require input space branching rather than ReLU branching; `verivital`, `mnistfc`, and `eran` benchmarks consist of

Table 2: VNN-COMP 2021 benchmarks: `oval21` and `cifar10-resnet`. Results marked with VNN-COMP are from publicly available benchmark data on VNN-COMP 2021 Github. "-" indicates unsupported model.

| Method | oval21 (30 properties; PGD upper bound 27) | | | cifar10-resnet (72 properties) | | |
|---|---|---|---|---|---|---|
| | time(s) | # verified | %timeout | time(s) | # verified | %timeout |
| nnenum* [2, 4] | 630.06 | 2 | 86.66 | - | - | - |
| Marabou* [31] | 429.13 | 5 | 73.33 | 157.70 | 39 | 45.83 |
| ERAN* [40, 38] | 233.84 | 6 | 70.00 | 129.48 | 43 | 40.28 |
| OVAL* [17, 16] | 393.14 | 11 | 53.33 | - | - | - |
| VeriNet* [25, 26] | 414.61 | 11 | 53.33 | 105.91 | 48 | 33.33 |
| $\alpha,\beta$-CROWN* [61, 60, 57] | 395.32 | 11 | 53.33 | 99.87 | 58 | 19.44 |
| MN-BaB [20] | 435.46 | 10 | 56.66 | - | - | - |
| Venus2† [7, 33] | 386.71 | 17 | 33.33 | - | - | - |
| $\alpha$-CROWN+MIP | 301.23 | 15 | 40.00 | 125.48 | 46 | 36.11 |
| GCP-CROWN with MIP cuts | **145.26** | **23** | **13.33** | **53.49** | **63** | **12.5** |

*  Results from VNN-COMP 2021 report [3].    † We use the latest code of Venus2 in the `vnncomp` branch, committed on Jul 18, 2022. Older versions cannot run these convolutional networks.

Table 3: **Verified accuracy (%)** and avg. per-example verification time (s) on 7 models from SDP-FO [15].

| Dataset $\epsilon = 0.3$ and $\epsilon = 2/255$ | Model | SDP-FO [15]* Verified% | Time (s) | PRIMA [40] Ver.% | Time(s) | $\beta$-CROWN [57] Ver.% | Time(s) | MN-BaB [20] Ver.% | Time(s) | Venus2 [7, 33] Ver.% | Time(s) | $\alpha$-CROWN+MIP Ver.% | Time(s) | GCP-CROWN Ver.% | Time(s) | Upper bound |
|---|---|---|---|---|---|---|---|---|---|---|---|---|---|---|---|---|
| MNIST | CNN-A-Adv | 43.4 | >20h | 44.5 | 135.9 | 70.5 | 21.1 | - | - | 35.5 | 148.4 | 56.5 | 224.3 | **72.0** | 19.9 | 76.5 |
| CIFAR | CNN-B-Adv | 32.8 | >25h | 38.0 | 343.6 | 46.5 | 32.2 | - | - | - | - | 27.0 | 360.6 | **48.5** | 57.8 | 65.0 |
| | CNN-B-Adv-4 | 46.0 | >25h | 53.5 | 43.8 | 54.0 | 11.6 | - | - | - | - | 52.5 | 129.5 | **59.0** | 21.5 | 63.5 |
| | CNN-A-Adv | 39.6 | >25h | 41.5 | 4.8 | 44.0 | 5.8 | 42.5 | 68.3 | 47.5 | 26.0 | 46.0 | 63.1 | **48.5** | 9.8 | 50.0 |
| | CNN-A-Adv-4 | 40.0 | >25h | 45.0 | 4.9 | 46.0 | 5.6 | 46.0 | 37.7 | 47.5 | 13.1 | **48.5** | 16.4 | **48.5** | 5.7 | 49.5 |
| | CNN-A-Mix | 39.6 | >25h | 37.5 | 34.3 | 41.5 | 49.6 | 35.0 | 140.3 | 33.5 | 72.4 | 32.5 | 231.3 | **47.5** | 29.2 | 53.0 |
| | CNN-A-Mix-4 | 47.8 | >25h | 48.5 | 7.0 | 50.5 | 5.9 | 49.0 | 70.9 | 49.0 | 37.3 | 52.5 | 77.7 | **55.5** | 12.4 | 57.5 |

*  We run $\alpha$-CROWN+MIP and MN-BaB with 600s timeout threshold for all models. "-" indicates that we could not run a model due to unsupported model structure or other errors. We run our GCP-CROWN with MIP cuts with a shorter 200s timeout for all models and it achieves better verified accuracy than all other baselines. Other results are reported from [57].

small MLP networks that can be solved directly by MIP solvers; `marabou` contains mostly adversarial examples, making it a good benchmark for falsifiers rather than verifiers. We present our results in Table 2. Besides results from 6 VNN-COMP 2021 participants, we also include two additional baselines, $\alpha$-CROWN+MIP (same as in Table 1), and MN-BaB [20], a recently proposed branch and bound framework with multi-neuron relaxations [39], which can be viewed as a restricted form of cutting planes. On the `oval21` benchmark, OVAL, VeriNet and $\alpha,\beta$-CROWN are the best performing tools, verified 11 out of 27 instances, while we can verify *twice more* instances (22 out of 27) on this benchmark. On the `cifar10-resnet` benchmark, our verifier also solves the most number of instances and achieves the lowest average time. In fact, $\alpha$-CROWN+MIP is also a strong baseline, solving 4 more instances than all competition participants, showing the importance of strong cutting planes. Our GCP-CROWN with MIP cuts combines the benefits of fast bound propagation on GPU with the strong cutting planes generated by a MIP solver and achieves the best performance. We present a more detailed analysis on the cutting planes used in this benchmark in Appendix C.2.

The `oval21` benchmark was also included as part of VNN-COMP 2022, concluded in July 2022, with a different sample of 30 instances. GCP-CROWN is part of the winning tool in VNN-COMP 2022, $\alpha,\beta$-CROWN, which verified 25 out of 30 instances in this benchmark, outperforming the second place tool (MN-BaB [20] with multi-neuron relaxations) with 19 verified instances by a large margin. More results on VNN-COMP 2022 can be found in these slides[1].

**Results on SDP-FO benchmarks.** We further evaluate our method on the SDP-FO benchmarks in [15, 57]. This benchmark contains 7 mostly adversarially trained MNIST and CIFAR models with 200 instances each, which are hard for many existing verifiers. Beyond the baselines reported in [57], we also include two additional baselines, $\alpha$-CROWN+MIP (same as in Table 1) and MN-BaB [20] (same as in Table 2). Table 3 shows that our method improves the percentage of verified images ("verified accuracy") on all models compared to state-of-the-art verifiers, further closing the gap between verified accuracy and empirical robust accuracy obtained by PGD attack (reported as "upper bound" in Table 3).

## 5 Related Work

Cutting plane method is a classic technique to strengthen the convex relaxation of an integer programming problem. Generic cutting planes such as Gomory's cut [21, 22], Chvátal–Gomory cut [12],

---

[1]A formal competition report is under preparation by VNN-COMP organizers; scores were presented in FLoC 2022: `https://drive.google.com/file/d/1nnRWSq3plsPvOT3V-drAF5D8zWGuO2VF/view`.

implied bound cut [27], lift-and-project [35], reformulation-linearization techniques [46] and mixed integer rounding cuts [41, 37] can be applied to almost any LP relaxed problems, and problem specific cutting planes such as Knapsack cut [14], Flow-cover cut [42] and Clique cut [29] require specific problem structures. Modern MIP solvers typically uses a branch-and-cut strategy, which tends to generate a large number of cuts before starting the next iteration of branching, and solve the LP relaxation of the MIP problem with cutting planes with an exact method such as the Simplex method. Our GCP-CROWN is a specialized solver for the NN verification problem, which can quickly obtain a lower bound of the LP relaxation with cutting planes specially for the NN verification problem.

The verification of piece-wise linear NNs can be formulated as a MIP problem, so early works [30, 31, 53] solve an integer or combinatorial formulation directly. For efficiency reasons, most recent works use a convex relaxation such as linear relaxation [19, 58] or semidefinite relaxation [44, 15]. Salman et al. [45] discussed the limitation of many convex relaxation based NN verification algorithms and coined the term "convex relaxation barrier", specifically for the popular single-neuron "Planet" relaxation [19]. Several works developed novel techniques to break this barrier. [47] added constraints that depends on the aggregation of multiple neurons, and these constraints were passed to a LP solver. [40] enhanced the multi-neuron formulation of [47] to obtain tighter relaxations. [1] studied stronger convex relaxations for a ReLU neuron after an affine layer, and [52] constructed efficient algorithms based on this relaxation for incomplete verification. [16] extended the formulation in [1] to a dual form and combined it with branch and bound to achieve completeness. [7] proposed specialized cuts by considering neuron dependencies and solve them using a MIP solver. [20] combined the multi-neuron relaxation [40] with branch and bound. Although these works can be seen as a special form of cutting planes, they mostly focused on enhancing the relaxation for several neurons within a single layer or two adjacent layers. GCP-CROWN can efficiently handle general cutting plane constraints with neurons from any layers in a bound propagation manner, and the cutting planes we find from a MIP solver can be seen as tighter convex relaxations encoding multi-neuron and multi-layer correlations.

## 6    Conclusion

In this paper, we propose GCP-CROWN, an efficient and GPU-accelerated bound propagation method for NN verification capable of handling any cutting plane constraints. We combine GCP-CROWN with branch and bound and high quality cutting planes generated by a MIP solver to tighten the convex relaxation for NN verification. The combination of fast bound propagation and strong cutting planes lead to state-of-the-art verification performance on multiple benchmarks. Our work opens up a great opportunity for studying more efficient and powerful cutting planes for NN verification.

**Limitations of this work**    Our work generalizes existing bound propagation methods that can handle only simple constraints (such as neuron split constraints in $\beta$-CROWN [57]) to general constraints, and we share a few common limitations as in previous works [10, 60, 16]: the branch-and-bound procces and bound propagation procedure are developed on ReLU networks, and it can be non-trivial to extend it to neural networks with non-piecewise-linear operations. In addition, we currently directly use cutting planes generated by a generic MIP solver, and there might exist stronger and faster cutting plane methods that can exploit the structure of the neural network verification problem. We hope these limitations can be addressed in future works.

**Potential Negative Societal Impact**    Our work focuses on formally proving desired properties of a neural network under investigation such as safety and robustness, which is an important direction of trustworthy machine learning and has overall positive societal impact. Since our verifier is a complete verifier, it might be possible to use it to find weakness of a neural network and guide adversarial attacks. However, we believe that formally characterizing a model's behavior and potenti al weakness is important for building robust models and preventing real-world malicious attack.

**Funding Disclosure**    This work is partially supported by the NSF grant No.1910100, NSF CNS No.2046726, NSF IIS No.2008173, NSF IIS No.2048280 and the Alfred P. Sloan Foundation. Huan Zhang is supported by a grant from the Bosch Center for Artificial Intelligence. Suman Jana acknowledges the NSF CAREER award.

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
