# Appendix

In Section A we derive the bound propagation procedure of GCP-CROWN and give a proof for Theorem 3.1 (GCP-CROWN bound propagation). In Section B we give additional background on branch and bound. In section C we give more details of experiments, more results as well as a case study for cutting planes used in GCP-CROWN with MIP cuts.

## A   The dual problem with cutting planes

In this section we derive the dual formulation for neural network verification problem with general cutting planes (or arbitrarily added linear constraints across any layers). Our derivation is based on the linearly relaxed MIP formulation with original binary variables $\mathbf{z}$ intact, to allow us to add cuts also to the integer variable (the mostly commonly used triangle relaxation does not have $\mathbf{z}$). We first write the full LP relaxation with arbitrary cutting planes, as well as their corresponding dual variables:

$$f_{\text{LP-cut}}^* = \min_{\boldsymbol{x}, \hat{\boldsymbol{x}}, \mathbf{z}} f(x)$$

$$\text{s.t.} \quad f(\boldsymbol{x}) = \boldsymbol{x}^{(L)}; \quad \boldsymbol{x}_0 - \epsilon \leq \boldsymbol{x} \leq \boldsymbol{x}_0 + \epsilon; \tag{2}$$

$$\boldsymbol{x}^{(i)} = \mathbf{W}^{(i)} \hat{\boldsymbol{x}}^{(i-1)} + \mathbf{b}^{(i)}; \quad i \in [L], \qquad \Rightarrow \boldsymbol{\nu}^{(i)} \in \mathbb{R}^{d_i} \tag{3}$$

$$\hat{x}_j^{(i)} \geq 0; \; j \in \mathcal{I}^{(i)} \qquad \Rightarrow \mu_j^{(i)} \in \mathbb{R} \tag{4}$$

$$\hat{x}_j^{(i)} \geq x_j^{(i)}; \; j \in \mathcal{I}^{(i)} \qquad \Rightarrow \tau_j^{(i)} \in \mathbb{R} \tag{5}$$

$$\hat{x}_j^{(i)} \leq u_j^{(i)} z_j^{(i)}; \; j \in \mathcal{I}^{(i)} \qquad \Rightarrow \gamma_j^{(i)} \in \mathbb{R} \tag{6}$$

$$\hat{x}_j^{(i)} \leq x_j^{(i)} - l_j^{(i)}(1 - z_j^{(i)}); \; j \in \mathcal{I}^{(i)} \qquad \Rightarrow \pi_j^{(i)} \in \mathbb{R} \tag{7}$$

$$\hat{x}_j^{(i)} = x_j^{(i)}; \; j \in \mathcal{I}^{+(i)} \tag{9}$$

$$\hat{x}_j^{(i)} = 0; \; j \in \mathcal{I}^{-(i)} \tag{10}$$

$$0 \leq z_j^{(i)} \leq 1; \quad j \in \mathcal{I}^{(i)}, \; i \in [L] \tag{11}$$

$$\sum_{i=1}^{L-1} \left( \boldsymbol{H}^{(i)} \boldsymbol{x}^{(i)} + \boldsymbol{G}^{(i)} \hat{\boldsymbol{x}}^{(i)} + \boldsymbol{Q}^{(i)} \mathbf{z}^{(i)} \right) \leq \boldsymbol{d} \qquad \Rightarrow \boldsymbol{\beta} \in \mathbb{R}^N \tag{15}$$

The Lagrangian function can be constructed as:

$$f_{\text{LP-cut}}^* = \min_{\boldsymbol{x}, \hat{\boldsymbol{x}}, \mathbf{z}} \max_{\boldsymbol{\nu}, \boldsymbol{\mu}, \boldsymbol{\tau}, \boldsymbol{\gamma}, \boldsymbol{\pi}, \boldsymbol{\beta}} \boldsymbol{x}^{(L)} + \sum_{i=1}^{L} \boldsymbol{\nu}^{(i)\top} \left( \boldsymbol{x}^{(i)} - \mathbf{W}^{(i)} \hat{\boldsymbol{x}}^{(i-1)} + \mathbf{b}^{(i)} \right)$$

$$+ \sum_{i=1}^{L-1} \sum_{j \in \mathcal{I}^{(i)}} \left[ \mu_j^{(i)}(-\hat{x}_j^{(i)}) + \tau_j^{(i)}(x_j^{(i)} - \hat{x}_j^{(i)}) + \gamma_j^{(i)}(\hat{x}_j^{(i)} - u_j^{(i)} z_j^{(i)}) + \pi_j^{(i)}(\hat{x}_j^{(i)} - x_j^{(i)} + l_j^{(i)} - l_j^{(i)} z_j^{(i)}) \right]$$

$$+ \boldsymbol{\beta}^\top \left[ \sum_{i=1}^{L-1} \left( \boldsymbol{H}^{(i)} \boldsymbol{x}^{(i)} + \boldsymbol{G}^{(i)} \hat{\boldsymbol{x}}^{(i)} + \boldsymbol{Q}^{(i)} \mathbf{z}^{(i)} \right) - \boldsymbol{d} \right]$$

$$\text{s.t.} \; \hat{x}_j^{(i)} = 0, j \in \mathcal{I}^{-(i)}; \quad \hat{x}_j^{(i)} = x_j^{(i)}, j \in \mathcal{I}^{+(i)}; \quad \boldsymbol{x}_0 - \epsilon \leq \boldsymbol{x} \leq \boldsymbol{x}_0 + \epsilon; \quad 0 \leq z_j^{(i)} \leq 1, j \in \mathcal{I}^{(i)}$$

$$\boldsymbol{\mu} \geq 0; \quad \boldsymbol{\tau} \geq 0; \quad \boldsymbol{\gamma} \geq 0; \quad \boldsymbol{\pi} \geq 0; \quad \boldsymbol{\beta} \geq 0$$

Here $\boldsymbol{\mu}, \boldsymbol{\tau}, \boldsymbol{\gamma}, \boldsymbol{\pi}$ are shorthands for all dual variables in each layer and each neuron. Note that for some constraints, their dual variables are not created because they are trivial to handle in the step steps. Rearrange the equation and swap the min and max (strong duality) gives us:

$$f_{\text{LP-cut}}^* = \max_{\boldsymbol{\nu},\boldsymbol{\mu},\boldsymbol{\tau},\boldsymbol{\gamma},\boldsymbol{\pi},\boldsymbol{\beta}} \min_{\boldsymbol{x},\hat{\boldsymbol{x}},\boldsymbol{z}} (\boldsymbol{\nu}^{(L)}+1)\boldsymbol{x}^{(L)} - \boldsymbol{\nu}^{(1)\top}\mathbf{W}^{(1)}\hat{\boldsymbol{x}}^{(0)}$$

$$+ \sum_{i=1}^{L-1} \sum_{j\in\mathcal{I}^{+(i)}} \left(\nu_j^{(i)} + \boldsymbol{\beta}^\top \boldsymbol{H}_{:,j}^{(i)} - \boldsymbol{\nu}^{(i+1)\top}\mathbf{W}_{:,j}^{(i+1)} + \boldsymbol{\beta}^\top \boldsymbol{G}_{:,j}^{(i)}\right) x_j^{(i)}$$

$$+ \sum_{i=1}^{L-1} \sum_{j\in\mathcal{I}^{-(i)}} \left(\nu_j^{(i)} + \boldsymbol{\beta}^\top \boldsymbol{H}_{:,j}^{(i)}\right) x_j^{(i)}$$

$$+ \sum_{i=1}^{L-1} \sum_{j\in\mathcal{I}^{(i)}} \Big[ \left(\nu_j^{(i)} + \boldsymbol{\beta}^\top \boldsymbol{H}_{:,j}^{(i)} + \tau_j^{(i)} - \pi_j^{(i)}\right) x_j^{(i)}$$

$$+ \left(-\boldsymbol{\nu}^{(i)\top}\mathbf{W}_{:,j}^{(i)} - \mu_j^{(i)} - \tau_j^{(i)} + \gamma_j^{(i)} + \pi_j^{(i)} + \boldsymbol{\beta}^\top \boldsymbol{G}_{:,j}^{(i)}\right) \hat{x}_j^{(i)}$$

$$+ \left(-u_j^{(i)}\gamma_j^{(i)} - l_j^{(i)}\pi_j^{(i)} + \boldsymbol{\beta}^\top \boldsymbol{Q}_{:,j}^{(i)}\right) z_j^{(i)} \Big]$$

$$- \sum_{i=1}^{L} \boldsymbol{\nu}^{(i)\top}\mathbf{b}^{(i)} + \sum_{i=1}^{L-1} \sum_{j\in\mathcal{I}^{(i)}} \pi_j^{(i)} l_j^{(i)} - \boldsymbol{\beta}^\top \boldsymbol{d}$$

$$\text{s.t.} \quad \boldsymbol{x}_0 - \epsilon \le \boldsymbol{x} \le \boldsymbol{x}_0 + \epsilon; \quad 0 \le z_j^{(i)} \le 1, j \in \mathcal{I}^{(i)}$$

$$\boldsymbol{\mu} \ge 0; \quad \boldsymbol{\tau} \ge 0; \quad \boldsymbol{\gamma} \ge 0; \quad \boldsymbol{\pi} \ge 0; \quad \boldsymbol{\beta} \ge 0$$

Here $\mathbf{W}_{:,j}^{(i+1)}$ denotes the $j$-th column of $\mathbf{W}^{(i+1)}$. Note that for the term involving $j \in \mathcal{I}^{+(i)}$ we have replaced $\hat{\boldsymbol{x}}^{(i)}$ with $\boldsymbol{x}^{(i)}$ to obtain the above equation. For the the term $x_j^{(i)}, j \in \mathcal{I}^{+(i)}$ it is always 0 so does not appear.

Solving the inner minimization gives us the dual formulation:

$$f_{\text{LP-cut}}^* = \max_{\boldsymbol{\nu},\boldsymbol{\mu},\boldsymbol{\tau},\boldsymbol{\gamma},\boldsymbol{\pi},\boldsymbol{\beta}} -\epsilon\|\boldsymbol{\nu}^{(1)\top}\mathbf{W}^{(1)}\boldsymbol{x}_0\|_1 - \sum_{i=1}^{L} \boldsymbol{\nu}^{(i)\top}\mathbf{b}^{(i)} - \boldsymbol{\beta}^\top \boldsymbol{d}$$

$$+ \sum_{i=1}^{L-1} \sum_{j\in\mathcal{I}^{(i)}} \left[\pi_j^{(i)} l_j^{(i)} - \text{ReLU}(u_j^{(i)}\gamma_j^{(i)} + l_j^{(i)}\pi_j^{(i)} - \boldsymbol{\beta}^\top \boldsymbol{Q}_{:,j}^{(i)})\right] \tag{17}$$

$$\text{s.t.} \quad \boldsymbol{\nu}^{(L)} = -1 \tag{18}$$

$$\nu_j^{(i)} = \boldsymbol{\nu}^{(i+1)\top}\mathbf{W}_{:,j}^{(i+1)} - \boldsymbol{\beta}^\top(\boldsymbol{H}_{:,j}^{(i)} + \boldsymbol{G}_{:,j}^{(i)}), \quad \text{for } j \in \mathcal{I}^{+(i)}, i \in [L-1] \tag{19}$$

$$\nu_j^{(i)} = -\boldsymbol{\beta}^\top \boldsymbol{H}_{:,j}^{(i)}, \quad \text{for } j \in \mathcal{I}^{-(i)}, i \in [L-1] \tag{20}$$

$$\nu_j^{(i)} = \pi_j^{(i)} - \tau_j^{(i)} - \boldsymbol{\beta}^\top \boldsymbol{H}_{:,j}^{(i)} \quad \text{for } j \in \mathcal{I}^{(i)}, i \in [L-1] \tag{21}$$

$$\left(\pi_j^{(i)} + \gamma_j^{(i)}\right) - \left(\mu_j^{(i)} + \tau_j^{(i)}\right) = \boldsymbol{\nu}^{(i+1)\top}\mathbf{W}_{:,j}^{(i+1)} - \boldsymbol{\beta}^\top \boldsymbol{G}_{:,j}^{(i)}, \quad \text{for } j \in \mathcal{I}^{(i)}, i \in [L-1] \tag{22}$$

$$\text{s.t.} \quad \boldsymbol{\mu} \ge 0; \quad \boldsymbol{\tau} \ge 0; \quad \boldsymbol{\gamma} \ge 0; \quad \boldsymbol{\pi} \ge 0; \quad \boldsymbol{\beta} \ge 0$$

The $\|\cdot\|_1$ comes from minimizing over $\boldsymbol{x}_0$ with the constraint $\boldsymbol{x}_0 - \epsilon \le \boldsymbol{x} \le \boldsymbol{x}_0 + \epsilon$, and the $\text{ReLU}(\cdot)$ term comes from minimizing over $z_j^{(i)}$ with the constraint $0 \le z_j^{(i)} \le 1$.

Before we give the bound propagation procedure, we first give a technical lemma:

**Lemma A.1.** *Given $u \ge 0$, $l \le 0$, $\pi \ge 0$, $\gamma \ge 0$, and $\pi + \gamma = C$, and define the function:*

$$g(\pi,\gamma) = -\text{ReLU}(u\gamma + l\pi + q) + l\pi$$

*Then*

$$\max_{\pi \geq 0, \gamma \geq 0} g(\pi, \gamma) = \begin{cases} l\pi^*, & \text{if } -uC \leq q \leq -lC \\ 0, & \text{if } q < -uC \\ -q, & \text{if } q > -lC \end{cases}$$

*where the optimal values for $\pi$ and $\gamma$ are:*

$$\pi^* = \max\left(\min\left(\frac{uC + q}{u - l}, C\right), 0\right), \quad \gamma^* = \max\left(\min\left(\frac{-lC - q}{u - l}, C\right), 0\right)$$

*Proof.* Case 1: when $u\gamma + l\pi + q \geq 0$, the objective becomes:

$$\max_{\pi \geq 0, \gamma \geq 0} g(\pi, \gamma) = -u\gamma - q$$
$$\text{s.t. } \pi + \gamma = C$$
$$u\gamma + l\pi + q \geq 0$$

Since $q$ is a constant and the object only involves a non-negative variable $\gamma$ with non-positive coefficient $-u$, $\gamma$ needs to be as smaller as possible as long as the constraint is satisfied. Substitute $\pi = C - \gamma$ into the constraint $u\gamma + l\pi + q \geq 0$,

$$u\gamma + l(C - \gamma) + q \geq 0 \Rightarrow \gamma \geq \frac{-lC - q}{u - l}$$

Considering $\gamma$ is within $[0, C]$, the optimal $\gamma$ and $\pi$ are:

$$\gamma^* = \max\left(\min\left(\frac{-lC - q}{u - l}, C\right)\right), \quad \pi^* = \max\left(\min\left(\frac{uC + q}{u - l}, C\right)\right)$$

Case 2: when $u\gamma + l\pi + q \leq 0$, the objective becomes:

$$\max_{\pi \geq 0, \gamma \geq 0} g(\pi, \gamma) = l\pi$$
$$\text{s.t. } \pi + \gamma = C$$
$$u\gamma + l\pi + q \leq 0$$

Since $q$ is a constant and the object only involves a non-negative variable $\pi$ with non-positive coefficient $l$, $\pi$ needs to be as smaller as possible as long as the constraint is satisfied. Substitute $\gamma = C - \pi$ into the constraint $u\gamma + l\pi + q \leq 0$,

$$u(C - \pi) + l\pi + q \leq 0 \Rightarrow \pi \geq \frac{uC + q}{u - l}$$

The optimal $\pi^*$ and $\gamma^*$ are the same as in case 1. The optimal objective can be obtained by substitute $\pi^*$ and $\gamma^*$ into $g(\pi, \gamma)$. The objective depends on $q$ because when $q$ is too large ($q \geq -lC$) or too small ($q \leq -uC$), $\pi^*$ and $\gamma^*$ are fixed at either 0 or $C$. $\square$

With this lemma, we are now ready to prove Theorem 3.1, the bound propagation rule with general cutting planes (GCP-CROWN):

**Theorem 3.1** (Bound propagation with general cutting planes)**.** *Given the following bound propagation rule on $\boldsymbol{\nu}$ with optimizable parameter $0 \leq \alpha_j^{(i)} \leq 1$ and $\boldsymbol{\beta} \geq 0$:*

$$\boldsymbol{\nu}^{(L)} = -1$$
$$\nu_j^{(i)} = \boldsymbol{\nu}^{(i+1)\top}\mathbf{W}_{:,j}^{(i+1)} - \boldsymbol{\beta}^\top(\boldsymbol{H}_{:,j}^{(i)} + \boldsymbol{G}_{:,j}^{(i)}), \qquad\qquad j \in \mathcal{I}^{+(i)}, \ i \in [L-1]$$
$$\nu_j^{(i)} = -\boldsymbol{\beta}^\top\boldsymbol{H}_{:,j}^{(i)}, \qquad\qquad j \in \mathcal{I}^{-(i)}, \ i \in [L-1]$$
$$\hat{\nu}_j^{(i)} := \boldsymbol{\nu}^{(i+1)\top}\mathbf{W}_{:,j}^{(i+1)} - \boldsymbol{\beta}^\top\boldsymbol{G}_{:,j}^{(i)} \qquad\qquad j \in \mathcal{I}^{(i)}, \ i \in [L-1]$$
$$\nu_j^{(i)} := \max\left(\min\left(\frac{u_j^{(i)}[\hat{\nu}_j^{(i)}]_+ + \boldsymbol{\beta}^\top\boldsymbol{Q}_{:,j}^{(i)}}{u_j^{(i)} - l_j^{(i)}}, [\hat{\nu}_j^{(i)}]_+\right), 0\right) - \alpha_j^{(i)}[\hat{\nu}_j^{(i)}]_- - \boldsymbol{\beta}^\top\boldsymbol{H}_{:,j}^{(i)} \qquad j \in \mathcal{I}^{(i)}, \ i \in [L-1]$$

Then $f_{LP\text{-}cut}^*$ is lower bounded by the following objective with any valid $0 \le \boldsymbol{\alpha} \le 1$ and $\boldsymbol{\beta} \ge 0$:

$$g(\boldsymbol{\alpha}, \boldsymbol{\beta}) = -\epsilon \|\boldsymbol{\nu}^{(1)\top} \mathbf{W}^{(1)} \boldsymbol{x}_0\|_1 - \sum_{i=1}^{L} \boldsymbol{\nu}^{(i)\top} \mathbf{b}^{(i)} - \boldsymbol{\beta}^\top \boldsymbol{d} + \sum_{i=1}^{L-1} \sum_{j \in \mathcal{I}^{(i)}} h_j^{(i)}(\boldsymbol{\beta})$$

where $h_j^{(i)}(\boldsymbol{\beta})$ is defined as:

$$h_j^{(i)}(\boldsymbol{\beta}) = \begin{cases} \dfrac{u_j^{(i)} l_j^{(i)} [\hat{\nu}_j^{(i)}]_+ + l_j^{(i)} \boldsymbol{\beta}^\top \boldsymbol{Q}_{:,j}^{(i)}}{u_j^{(i)} - l_j^{(i)}} & \text{if } l_j^{(i)} [\hat{\nu}_j^{(i)}]_+ \le \boldsymbol{\beta}^\top \boldsymbol{Q}_{:,j}^{(i)} \le u_j^{(i)} [\hat{\nu}_j^{(i)}]_+ \\ 0 & \text{if } \boldsymbol{\beta}^\top \boldsymbol{Q}_{:,j}^{(i)} \ge u_j^{(i)} [\hat{\nu}_j^{(i)}]_+ \\ \boldsymbol{\beta}^\top \boldsymbol{Q}_{:,j}^{(i)} & \text{if } \boldsymbol{\beta}^\top \boldsymbol{Q}_{:,j}^{(i)} \le l_j^{(i)} [\hat{\nu}_j^{(i)}]_+ \end{cases}$$

*Proof.* Given (22), for $j \in \mathcal{I}^{(i)}$, observing that the upper and lower bounds of ReLU relaxations for a single neuron cannot be tight simultaneously [58], $\pi_j^{(i)} + \gamma_j^{(i)}$ and $\mu_j^{(i)} + \tau_j^{(i)}$ cannot be both non-zero[2]. Thus, we have

$$\pi_j^{(i)} + \gamma_j^{(i)} = \left[\boldsymbol{\nu}^{(i+1)\top} \mathbf{W}_{:,j}^{(i+1)} - \boldsymbol{\beta}^\top \boldsymbol{G}_{:,j}^{(i)}\right]_+ := \left[\hat{\nu}_j^{(i)}\right]_+, \tag{23}$$

$$\mu_j^{(i)} + \tau_j^{(i)} = \left[\boldsymbol{\nu}^{(i+1)\top} \mathbf{W}_{:,j}^{(i+1)} - \boldsymbol{\beta}^\top \boldsymbol{G}_{:,j}^{(i)}\right]_- := \left[\hat{\nu}_j^{(i)}\right]_-. \tag{24}$$

To avoid clutter, we define $\hat{\nu}_j^{(i)} := \boldsymbol{\nu}^{(i+1)\top} \mathbf{W}_{:,j}^{(i+1)} - \boldsymbol{\beta}^\top \boldsymbol{G}_{:,j}^{(i)}$, and we define $[\,\cdot\,]_+ := \max(0, \cdot)$ and $[\,\cdot\,]_- := \min(0, \cdot)$.

To derived the proposed bound propagation rule, in (17), we must eliminate variable $\gamma_j^{(i)}$ and $\pi_j^{(i)}$. Ignoring terms related to $\boldsymbol{\nu}$, we observe that we can optimize the term $\pi_j^{(i)} l_j^{(i)} - \text{ReLU}(u_j^{(i)} \gamma_j^{(i)} + l_j^{(i)} \pi_j^{(i)} - \boldsymbol{\beta}^\top \boldsymbol{Q}_{:,j}^{(i)})$ for each $j$ for $\mathcal{I}^{(i)}$ individually. We seek the optimal solution for the follow optimization problem on function $h$:

$$\max_{\pi_j^{(i)} \ge 0, \gamma_j^{(i)} \ge 0} h_j^{(i)}(\pi_j^{(i)}, \gamma_j^{(i)}; \boldsymbol{\beta}) := \pi_j^{(i)} l_j^{(i)} - \text{ReLU}(u_j^{(i)} \gamma_j^{(i)} + l_j^{(i)} \pi_j^{(i)} - \boldsymbol{\beta}^\top \boldsymbol{Q}_{:,j}^{(i)}) \tag{25}$$

$$\text{s.t. } \pi_j^{(i)} + \gamma_j^{(i)} = \left[\hat{\nu}_j^{(i)}\right]_+$$

$$\pi_j^{(i)} \ge 0; \quad \gamma_j^{(i)} \ge 0$$

Note that we optimize over $\pi_j^{(i)}$ and $\gamma_j^{(i)}$ here and treat $\boldsymbol{\beta}$ as a constant. Applying Lemma A.1 with $\pi = \pi_j^{(i)}, \gamma = \gamma_j^{(i)}, u = u_j^{(i)}, l = l_j^{(i)}, q = -\boldsymbol{\beta}^\top \boldsymbol{Q}_{:,j}^{(i)}, C = [\hat{\nu}_j^{(i)}]_+$ we obtain the optimal objective for (25):

$$h_j^{(i)}(\boldsymbol{\beta}) = \begin{cases} l_j^{(i)} \pi_j^{(i)} & \text{if } l_j^{(i)} [\hat{\nu}_j^{(i)}]_+ \le \boldsymbol{\beta}^\top \boldsymbol{Q}_{:,j}^{(i)} \le u_j^{(i)} [\hat{\nu}_j^{(i)}]_+ \\ 0 & \text{if } \boldsymbol{\beta}^\top \boldsymbol{Q}_{:,j}^{(i)} \ge u_j^{(i)} [\hat{\nu}_j^{(i)}]_+ \\ \boldsymbol{\beta}^\top \boldsymbol{Q}_{:,j}^{(i)} & \text{if } \boldsymbol{\beta}^\top \boldsymbol{Q}_{:,j}^{(i)} \le l_j^{(i)} [\hat{\nu}_j^{(i)}]_+ \end{cases}, \quad \pi_j^{(i)} = \max\left(\min\left(\frac{u_j^{(i)} [\hat{\nu}_j^{(i)}]_+ + \boldsymbol{\beta}^\top \boldsymbol{Q}_{:,j}^{(i)}}{u_j^{(i)} - l_j^{(i)}}, C\right), 0\right)$$

Substitute this solution of $\pi_j^{(i)}$ into (21), and also observe that due to (24), $\tau_j^{(i)}$ is a variable between $0$ and $\left[\hat{\nu}_j^{(i)}\right]_-$, so we can rewrite (21) as:

$$\nu_j^{(i)} := \max\left(\min\left(\frac{u_j^{(i)} [\hat{\nu}_j^{(i)}]_+ + \boldsymbol{\beta}^\top \boldsymbol{Q}_{:,j}^{(i)}}{u_j^{(i)} - l_j^{(i)}}, [\hat{\nu}_j^{(i)}]_+\right), 0\right) - \alpha_j^{(i)} [\hat{\nu}_j^{(i)}]_- - \boldsymbol{\beta}^\top \boldsymbol{H}_{:,j}^{(i)}$$

---

[2]In theorey, it is possible that $\tau_j^{(i)} \ne 0, \gamma_j^{(i)} \ne 0, \pi_j^{(i)} = \mu_j^{(i)} = 0$ or $\pi_j^{(i)} \ne 0, \mu_j^{(i)} \ne 0, \tau_j^{(i)} = \gamma_j^{(i)} = 0$. This situation may happen when $u_j^{(i)}$ or $l_j^{(i)}$ is exactly tight and the solution $x_j^{(i)}$ is at the intersection of one lower and upper linear equatlities. Practically, this can be avoided by adding a small $\delta$ to $u_j^{(i)}$ or $l_j^{(i)}$, and in most scenarios $u_j^{(i)}$ are $l_j^{(i)}$ are loose bounds so this situation will not occur. The same argument is also applicable to [58].

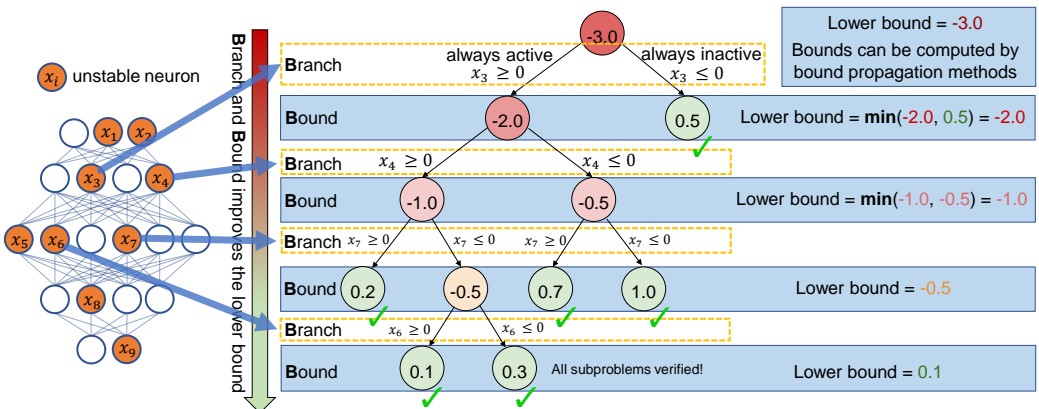

Figure 3: **B**ranch and **B**ound (BaB) for neural network verification. A *unstable* ReLU neuron (e.g., $x_1$ may receive either negative or positive inputs for some input $\boldsymbol{x}$ in perturbation set $\mathcal{C}$. On the other hand, a stable ReLU neuron is always positive or negative for all $\boldsymbol{x} \in \mathcal{C}$, so it is a linear operation and branching is not needed.

Here $0 \leq \alpha_j^{(i)} \leq 1$ is an optimizable parameter that can be updated via its gradient during bound propagation, and any $0 \leq \alpha_j^{(i)} \leq 1$ produces valid lower bounds. All above substitutions replace each dual variable $\boldsymbol{\pi}$, $\boldsymbol{\gamma}$ and $\boldsymbol{\mu}$ to a valid setting in the dual formulation, so the final objective $g(\boldsymbol{\alpha}, \boldsymbol{\beta})$ is always a sound lower bound of $f^*_{\text{LP-cut}}$. This theorem establishes the soundness of GCP-CROWN. $\qquad \square$

Note that an optimal setting of $\boldsymbol{\alpha}$ and $\boldsymbol{\beta}$ do not necessarily lead to the optimal primal value $f^*_{\text{LP-cut}}$. The main reason is that (25) gives a closed-form solution for $\boldsymbol{\pi}$ and $\boldsymbol{\gamma}$ to eliminate these variables without considering other dual variables like $\boldsymbol{\nu}$ to simplify the bound propagation process. To achieve theoretical optimality, $\boldsymbol{\pi}$ and $\boldsymbol{\gamma}$ can also be optimized.

## B   More technical details and background of GCP-CROWN with MIP cuts

**Branch-and-bound**   Our work is based on branch-and-bound (BaB), a powerful framework for neural network verification [10] which many state-of-the-art verifiers are based on [57, 16, 26, 20]. Here we give a brief introduction for the concept of branch-and-bound for readers who are not familiar with this field.

We illustrate the branch-and-bound process in Figure 3. Neural network verification seeks to minimize the objective (1): $f^* = \min_{\boldsymbol{x}} f(\boldsymbol{x}), \forall \boldsymbol{x} \in \mathcal{C}$, where $f(\boldsymbol{x})$ is a ReLU network under our setting. A positive $f^*$ indicates that the network can be verified. However, since ReLU neurons are non-linear, this optimization problem is non-convex, and it usually requires us to relax ReLU neurons with linear constraints (e.g., the LP relaxation in (11)) to obtain a lower bound for objective $f^*$. This lower bound ($-3.0$ at the root node in Figure 3) might become negative, even if $f^*$ (the unknown ground-truth) is positive. Branch-and-bound is a systematic method to tighten this lower bound. First, we split an unstable ReLU neuron, such as $x_3$, into two cases: $x_3 \geq 0$ and $x_3 \leq 0$ (the *branching step*), producing two subproblems each with an additional constraint $x_3 \geq 0$ or $x_3 \leq 0$. In each subproblem, neuron $x_3$ does not need to be relaxed anymore, so the lower bound of $f^*$ becomes tighter in each of the two subdomains in the subsequent *bounding step* ($-3.0$ becomes $-2.0$ and $0.5$) in Figure 3. Any subproblem with a positive lower bound is successfully verified and no further split is needed. Then, we repeat the branching and bounding steps on subproblems that still have a negative lower bound. We terminate when all unstable neurons are split or all subproblems are verified. In Figure 3, all leaf subproblems have positive lower bounds so the network is verified. On the other hand, if we have split all unstable neurons and there still exist domains with negative bounds, a counter-example can be constructed.

The contribution of this work is to improve the bounding step using cutting planes. As one can observe in Figure 3, if we can make the bounds tighter, they approach to zero faster and less branching is needed. Since the number of the subproblems may be exponential to the number of unstable neurons

in the worst case, it is crucial to improve the bounds quickly. Informally, cutting planes are additional valid constraints for the relaxed subproblems, and adding these additional constraints as in (16) makes the lower bounds tighter. We refer the readers to integer programming literature [5, 13] for more details on cutting plane methods. Existing neural network verifiers mostly use bound propagation method [61, 50, 60, 57, 16] for the bounding step thanks to their efficiency and scalability, but none of them can handle general cutting planes. Our GCP-CROWN is the first bound propagation method that supports general cutting planes, and we demonstrated its effectiveness in Section 4 when combined with cutting planes generated by a MIP solver.

**Soundness and Completeness**  GCP-CROWN is sound because Theorem 3.1 guarantees that we always obtain a sound lower bound of the verification problem as long as valid cutting planes (generated by a MIP solver in our case) are added. Our method, when combined with branch-and-bound, is also complete because we provide a strict improvement in the bounding step over existing methods such as [57], and the arguments for completeness in existing verifiers using branch-and-bound on splitting ReLU neurons such as [9, 57] are still valid for this work.

# C   More details on experiments

## C.1   Experimental Setup

Our experiments are conducted on a desktop with an AMD Ryzen 9 5950X CPU, one NVIDIA RTX 3090 GPU (24GB GPU memory), and 64GB CPU memory. Our implementation is based on the open-source $\alpha,\beta$-CROWN verifier[3] with cutting plane related code added. Both $\alpha$-CROWN+MIP and GCP-CROWN with MIP cuts use all 16 CPU cores and 1 GPU. `gurobi` is used as the MIP solver in $\alpha$-CROWN+MIP. Although `gurobi` usually outperforms other MIP solvers for NN verification problems, it cannot export cutting planes, so our cutting planes are acquired by the `cplex` [28] solver (version 22.1.0.0). We use the Adam optimizer [32] to solve both $\alpha$ and $\beta$ with 20 iterations. The learning rates are set as 0.1 and 0.02 (0.01 for `oval20`) for optimizing $\alpha$ and $\beta$ respectively. We decay the learning rates with a factor of 0.9 (0.8 for `oval20`) per iteration. Timeout for properties in `oval20` are set as 1 hour follow the original benchmark. Timeout for `oval21` and `cifar10-resnet` are set as 720s and 300s respectively, the same as in VNN-COMP 2021. Timeout for SDP-FO models are set as 600s for $\alpha$-CROWN+MIP and MN-BaB, and a shorter 200s timeout is used for GCP-CROWN with MIP cuts.

We summarize the model structures and batch size used in our experiments in Table 4. The CIFAR-10 Base, Wide and Deep models are used in `oval20` and `oval21` benchmarks.

Table 4: Model structures used in our experiments. The notation Conv($a$, $b$, $c$) stands for a conventional layer with $a$ input channel, $b$ output channels and a kernel size of $c \times c$. Linear($a$, $b$) stands for a fully connected layer with $a$ input features and $b$ output features. ResBlock($a$, $b$) stands for a residual block that has $a$ input channels and $b$ output channels. We have ReLU activation functions between two consecutive linear or convolutional layers.

| Model name | Model structure | Batch size |
|---|---|---|
| Base (CIFAR-10) | Conv(3, 8, 4) - Conv(8, 16, 4) - Linear(1024, 100) - Linear(100, 10) | 1024 |
| Wide (CIFAR-10) | Conv(3, 16, 4) - Conv(16, 32, 4) - Linear(2048, 100) - Linear(100, 10) | 1024 |
| Deep (CIFAR-10) | Conv(3, 8, 4) - Conv(8, 8, 3) - Conv(8, 8, 3) - Conv(8, 8, 4) - Linear(412, 100) - Linear(100, 10) | 1024 |
| `cifar10-resnet2b` | Conv(3, 8, 3) - ResBlock(8, 16) - ResBlock(16, 16)) - Linear(1024, 100) - Linear(100, 10) | 2048 |
| `cifar10-resnet4b` | Conv(3, 16, 3) - ResBlock(16, 32) - ResBlock(32, 32)) - Linear(512, 100) - Linear(100, 10) | 2048 |
| CNN-A-Adv (MNIST) | Conv(1, 16, 4) - Conv(16, 32, 4) - Linear(1568, 100) - Linear(100, 10) | 4096 |
| CNN-A-Adv/-4 (CIFAR-10) | Conv(3, 16, 4) - Conv(16, 32, 4) - Linear(2048, 100) - Linear(100, 10) | 4096 |
| CNN-B-Adv/-4 (CIFAR-10) | Conv(3, 32, 5) - Conv(32, 128, 4) - Linear(8192, 250) - Linear(250, 10) | 1024 |
| CNN-A-Mix/-4 (CIFAR-10) | Conv(3, 16, 4) - Conv(16, 32, 4) - Linear(2048, 100) - Linear(100, 10) | 4096 |

## C.2   A case study on cutting planes

The effectiveness of GCP-CROWN with MIP cuts motivates us to take a more careful look at the cutting planes generated by off-the-shelf MIP solvers, and understand how well it can contribute to

---

[3]`https://github.com/huanzhang12/alpha-beta-CROWN`

tighten the lower bounds and strengthen verification performance. We use the `oval21` as a sample case study because GCP-CROWN with MIP cuts is very effective on this benchmark.

The `oval21` benchmark has 30 instances, each with 9 target labels (properties) to verify. Among the total of 270 properties, we filter out the easy cases where fast incomplete verifiers like $\alpha$-CROWN can verify directly, and 39 *hard properties* remain which must be solved using branch and bound and/or cutting planes. Cutting planes are generated on these hard properties and they greatly help GCP-CROWN.

**Number of cuts used to verify each property.** The maximal number of cuts applied per property is 4,162 and the minimal number is 318. On average, we have 1,683 cuts applied to our GCP-CROWN to solve these hard properties.

**Improvements on lower bounds.** In branch-and-bound, we lower bound the objective of Eq. 1 with ReLU split constraints [57]. A tighter lower bound can reduce the number of branches used and usually leads to stronger verification. We measure how well the cuts generated by off-the-shelf solvers in improving the tightness of the lower bound for verification. Without branching and cutting planes, the average $\alpha$-CROWN lower bound is -2.54. With generated cutting planes, we can improve the lower bound by 0.51 on average and can directly verify 4 out of 39 hard properties without branching. The lower bound without branching can be maximally improved by 1.54 and minimally improved by 0.04.

**Structure of Generated Cuts.** Finally, we investigate the variables involved in each generated cutting plane. In total, the MIP solver generates 65,647 cutting planes in total for the 39 hard properties. 65,301 of the cuts involve variables across multiple layers. It indicates that single layer cutting planes (e.g., constraints involving pre- and post-activation variables of a single ReLU layer only) commonly used in previous works [47, 40] are not optimal in general. Additionally, all of 65,647 cuts have at least one ReLU *integer variable* $\mathbf{z}^{(i)}$ used, which was not supported in existing bound propagation methods. 65,197 of all cuts involve at least one variable of input neurons. 23,600 of all cuts have at least one pre-ReLU variables and 51,617 of them have at least one post-ReLU variables.