# OpenReview forum: "General Cutting Planes for Bound-Propagation-Based Neural Network Verification"
_NeurIPS.cc/2022/Conference — NeurIPS 2022 Accept_

### Official Review · Reviewer_V4Wg · 2022-07-07

**Rating:** 6
**Confidence:** 5
**Soundness:** 3 good
**Presentation:** 1 poor
**Contribution:** 3 good

**Summary:**

The authors extend the popular and effective Beta-CROWN algorithm, a complete verifier that very efficiently operates on the triangle ReLU relaxation in the context of branch and bound, to support general cutting planes.
The presented algorithm, named GCP-CROWN, relies on cutting planes from off-the-shelf LP solvers, run on a CPU, in parallel to the GPU-based neural network verification procedure.
GCP-CROWN is shown to outperform state-of-the-art baselines on relevant benchmarks.

**Questions:**

- I believe MN-BaB, which is closely related to GCP-CROWN, should be more accurately described and presented in the technical part of the paper;

- the authors should explain and argue more on why one has to use these cutting planes in the context of bound propagation rather than lagrangian relaxation or decomposition, where doing so would be easier. It would be nice to see an experimental comparison on that, or theoretical arguments on why bound propagation is to be preferred. From the technical parts of the paper, it would seem like bound propagation methods are the only GPU-friendly alternative to off-the-shelf LP solvers. Many cited works [6, 13, 15] demonstrate otherwise, and that should be more carefully explained.

- Theorem 3.1 shows that GCP-CROWN provides a lower bound to the LP with cutting planes. However, do the optimal values of alpha and beta lead to the optimal primal value?

- what's the employed branching strategy in the experiments?

- why does GCP-CROWN take more subproblems than beta-CROWN on the deep network of table 1? That's counter-intuitive.

**Limitations:**

The authors adequately address the limitations of their work, but should be more upfront with respect to their incremental improvements on beta-CROWN.

**Strengths And Weaknesses:**

While the idea behind GCP-CROWN is simple and somewhat incremental (it is a minor, yet powerful, extension of beta-CROWN), it is undeniably novel in the context of neural network verification. Furthermore, the experimental results clearly indicate that GCP-CROWN outperforms the state of the art on hard verification benchmarks.

My concerns mostly revolve around the sub-par quality of the presentation. Crucially, I believe that the authors should be more upfront with respect to their algorithm's incrementality with respect to beta-CROWN, which is not necessarily a bad thing. However, such incrementality might not be evident to someone outside the sub-field. In other words, the presentation should start from beta-CROWN, and then move on to include the relaxed integer variables, and the cutting planes.
The analysis of the cutting planes, presented in the appendix, is perhaps worth including in the main paper.
Furthermore, it would be nice if the authors could include larger networks in the evaluations: for instance, the cifar2020 benchmark from VNN-COMP-21 includes networks with tens of thousands of activations. I would be curious to see whether cutting planes still pay off with respect to beta-CROWN at that scale.

---

> ### Author Response · Authors · 2022-08-02
> **We thank the reviewer for recognizing the importance of our paper and we have revised our paper and answered questions below (Part 1)**
>
> We greatly appreciate the constructive feedback from the reviewer, and thank you for recognizing the importance of this work for the field of neural network verification. We revised our paper to highlight more connections to beta-CROWN earlier as well as discussing other GPU accelerated methods such as Lagrangian decomposition. We address each of your questions below:
>
> > Weakness 1: Presentation should start with beta-CROWN
>
> We thank you for the suggestion on the paper presentation. We took your suggestion in our revision and mentioned the connection to previous works earlier at the beginning of section 3; we also considered the option of starting with the derivation of (beta-)CROWN when writing the paper; however, one challenge is that following Section 3.1 in Wang et al. 2021 there is no intuitive way to introduce the integer variables as they do not naturally appear at all in the formulation of classical bound propagation (i.e., checking the signs of the coefficient matrices and take the lower/upper bounds accordingly). We also apologize if the presentation seemed like it hid the connection to previous works - we explicitly discussed the relationship among GCP-CROWN, convex outer adversarial polytope, and beta-CROWN at the end of section 3.1 which should hopefully make it evident to readers (we added a few sentence in section 3 to make the connection even more evident).  Hopefully these changes address your concerns in this regard.
>
> > Weakness 2: incrementality with respect to beta-CROWN
>
> Although we agree with the reviewer that our well has a close relationship to beta-CROWN and other bound propagation methods, which are special cases of ours, **the technical changes we made to existing bound propagation like beta-CROWN are not trivial**. The three main technical contributions include (1) traditional bound propagation only involves variables from two consecutive layers, but our cutting plane is general can involve variables from any layers, which is conceptually not straightforward and initially we thought it would be impossible to do it in a propagation manner; (2) the addition of integer variables that do not exist in existing bound propagation formulations, which can be very useful to encode cutting planes, and (3) we demonstrated experimentally that the addition of these components in bound propagation is highly effective and efficient and can solve hard problems that none of existing verifiers can solve.
>
> > Weakness 3: cifar2020 benchmark from VNN-COMP-21
>
> We tried the cifar2020 networks as suggested, however the issue with this benchmark is most instances are simple (because they were trained using a certified defense) and can already be solved using beta-CROWN quickly, and the room for improvements is little (only 13 out of 203 instances remain unsolved). We were able to generate cutting planes on these models. However, we cannot solve any additional instances on this particular benchmark. In fact, in the recently concluded VNN-COMP 2022, most state-of-the-art  tools are evaluated on this benchmark again but none of them was able to verify a single additional instance (see [VNN-COMP 2022 presentation, page 99](https://drive.google.com/file/d/1nnRWSq3plsPvOT3V-drAF5D8zWGu02VF/view)).
>
> > Weakness 4: Scalability of cutting planes for large models
>
> The scalability of our general bound propagation procedure is quite good - adding thousands cutting planes on a relatively large ResNet model takes only 1.5x - 2x more propagation time compared to beta-CROWN and is overall beneficial due to the tighter bounds (e.g., in the cifar10_resnet benchmark in Table 2, and we also recently tested on even larger models from [VNN-COMP 2022](https://drive.google.com/file/d/1nnRWSq3plsPvOT3V-drAF5D8zWGu02VF/view)). Our paper showcases the possibility of using cutting planes generated by a MILP solver so for this particular type of cutting planes, its scalability is limited by a MILP solver; however since we do not rely on a MILP solver to solve the entire problem and it can be run in the background, its scalability is still better than traditional approaches of using a MILP solver alone (e.g., it outperforms other baselines on cifar10-resnet in Table 2, one of the largest models in VNN-COMP 2021 scoring benchmarks). How to efficiently generate effective cutting planes for even larger networks is still an open challenge and cannot be addressed in a single paper. **Our work opens up entirely new opportunities to solve hard verification problems, not just limited to the MIP solver based cutting planes discussed in our paper**.
>
>
> > Weakness 4: Analysis of the cutting planes
>
> We will move it to the main paper in the final version, where one additional page is allowed.

---

> > ### Author Response · Authors · 2022-08-02
> > **We thank the reviewer for recognizing the importance of our paper and we have revised our paper and answered questions below (Part 2)**
> >
> > > Question 1: Description of MN-BaB
> >
> > We added a discussion in Section 3.1 on MN-BaB, which is similar in many respects to beta-CROWN.  But unlike our work, they cannot handle general cutting planes such as those including integer variables and cross-layer general cutting planes.
> >
> > > Question 2: Cutting planes in the context of lagrangian relaxation or decomposition
> >
> > We believe cutting planes are also possible with Lagrangian relaxation or decomposition based approaches, although in our paper we derived it from the bound propagation perspective. Frankly, many of these approaches share the same root and we believe they all can be extended to general cutting planes, and we have no strong opinion one is better than the others (we added a short discussion on this in Section 3.1). **We also fixed the misleading sentences** in section 1 and 3 that seem to claim bound propagation methods are the only GPU-friendly alternative. Please check the revision and if you find any potentially misleading sentence you are unhappy with, please let us know.
> >
> > > Question 3: Optimal values of alpha and beta
> >
> > An optimal setting of alpha and beta do not always lead to the optimal primal value. The main reason is that Eq. (25) gives a closed-form solution for $\pi$ and $\gamma$ to eliminate these variables without considering other dual variables like $\nu$. To achieve theoretical optimality, $\pi$ and $\gamma$ can also be optimized. Practically we found that on real problems we get very close to the optimal primal value and creating additional optimizable variables make the algorithm unnecessarily complex and slower. We added this discussion to the Appendix.
> >
> > > Question 4: Branching strategy
> >
> > We use filtered smart branching (FSB) in all experiments. We mentioned this method and cited the paper (De Palma et al., 2021) in our revision.
> >
> > > Question 5: More subproblems in deep networks
> >
> > It is a minor implementation difference. In our branch and bound implementation, we sometimes proactively split more than one ReLU neuron at once so it may create unnecessary branches for very simple instances (which is the case for most points in the deep model). For harder models such as the base and wide models, the impact on the reported number of branches is minimal.
> >
> > We’ve uploaded a revised version of our paper, but due to page limits we had to keep the additions very short (highlighted in blue). If our paper is accepted, we will have one additional page to further extend the discussions requested by the reviewer. We hope the reviewer can re-evaluate our paper based on our response. Thank you.

---

> ### Author Response · Authors · 2022-08-08
> **Thank you again for your constructive feedback, and hope to hear from you before the discussion period is closed**
>
> Dear V4Wg,
>
> Thank you again for recognizing the importance of our work in the field of neural network verification, and give many detailed suggestions. Based on your comments, we have revised our paper to highlight connections to beta-CROWN clearly, added a discussion of MN-BaB, and cited and discussed Lagrangian decomposition based approaches. Due to page limits, we had to keep the additions very short (highlighted in blue), but in the final revision, we will take the additional page for an extended discussion.
>
> We also answered your questions in detail in our response. Because the discussion period is closing soon, we hope you can take a look at our response, and please kindly let us know if you have any additional questions or comments. Thank you.
>
> Sincerely,
> Paper 9785 Authors

---

> ### Author Response · Authors · 2022-08-09
> **We really appreciate your constructive feedback and please let us know if you have any additional questions before the discussion period is closed**
>
> Dear Reviewer V4Wg,
>
> We greatly appreciate your detailed and constructive feedback! Your suggestions are very helpful to us and we revised our writing and discussed new papers as suggested. We also answered each question in detail in our response, and will extend relevant discussions into our final version of the paper. Please kindly let us know if you have any additional questions during the last a few hours of discussion period, and we hope you can positively support our paper in the final discussion with the AC. Thank you!
>
> Sincerely,
> Paper 9785 Authors

---

### Official Review · Reviewer_QMej · 2022-07-11

**Rating:** 7
**Confidence:** 4
**Soundness:** 3 good
**Presentation:** 2 fair
**Contribution:** 3 good

**Summary:**

These paper improve the general BaB method by integrateing any cutting cut method for formal complete evrifcation of adervarial robustness. This computation allows parallel comptation . they outperfom state of the art againstal several datset, methods, noise level for multilpes models for both verified accuracy and verification time.

**Questions:**

Can you compare the power ressources used compared to methods DP-FO [12]∗ PRIMA [35] β-CROWN [52] MN-BaB [17] α-CROWN+MIP ? (as you used parralel comptation: what if you did not use paralilsum)?

You did not discuss the limitations of your work

You did not dscuss the comprarsion of different cutting planes.

**Limitations:**

cf above

**Strengths And Weaknesses:**

Strengths:

Novelty of the approach: complete verification of DNN specific family

Important topic

Sound proof of the Hessian bounds

Clarity of the paper

Faster  verification and higher verified accuracy

Weaknesses

Poor experiments comparison according power resoureces

---

> ### Author Response · Authors · 2022-08-02
> **Thank you for the positive feedback and we answer your questions below**
>
> We sincerely thank the reviewer for recognizing the importance and contribution of our work and giving us encouraging comments. We answer your questions below:
>
> > Question 1. Computation resource used in experiments
>
> All algorithms compared here utilize a single GPU, although they may or may not be able to take advantage of additional CPU cores.
> - SDP-FO and beta-CROWN utilize a single RTX 3090 GPU and their algorithms cannot take advantage of more than one CPU core.
> - PRIMA and MN-BAB utilize both CPUs and GPUs. MN-BaB experiments were conducted on a machine with 16 CPU cores and an RTX 3090 GPU. PRIMA results were from (Wang et al., 2021) which used 20 CPU cores with 1 GPU.
> - α-CROWN+MIP: the α-CROWN bounds are computed on a single RTX 3090 GPU, the MIP solver is then run on a 16-core CPU.The two parts cannot run in parallel because MIP depends on the α-CROWN bounds.
> - GCP-CROWN (ours): we use a single RTX 3090 GPU with a single CPU core to run bound propagation and branch and bound. The remaining 15 CPU cores run MIP solvers in parallel to find cutting planes.
>
> > Question 2. Parallelism in our algorithm and comparisons in experiments
>
> Parallelism is natural in our setting because the MIP solver runs on CPUs to find cutting planes, and the bound-propagation-based branch and bound solver runs on a GPU independently. We believe the comparison is fair because it is a unique benefit of our algorithm to use both types of hardware. Most other algorithms cannot take advantage of both GPU and CPU, even if they run on the same machine.  We thus feel that wall clock time (using the computational resources we describe) to be the best method for evaluating the real benefit of our method.
>
> > Question 3. Limitations of this work
>
> We discussed limitations and societal impacts of this work at the end of Appendix B (supplementary material). The main limitation (also shared by many previous works) is that the branch and bound procedure in our work is currently limited to ReLU networks. Additionally, there could be more powerful methods to find cutting planes compared to using a MILP solver.
>
> > Question 4. Comparison of different cutting planes
>
> Our work is the first work considering general cutting planes for neural network verification using an efficient bound propagation method, so there is no direct baseline using general cutting planes to compare against. In our experiments, we compare to MN-BaB which can be seen as a special case of cutting planes. Additionally, during the response period, we also added a comparison to Venus2 (E. Botoeva et al.) which also involves a specialized cutting plane method. We outperform both approaches. In addition, we conducted a detailed analysis of cutting planes used in our paper in Appendix C.2.
>
> We thank you again for your support of our paper and we hope our answers are helpful.

---

> ### Author Response · Authors · 2022-08-09
> **Thank you again for your encouraging comments! Let us know if you have any additional questions before the discussion period ends.**
>
> Dear Reviewer QMej,
>
> We greatly appreciate your encouraging review and thank you again for recognizing the importance of our work! We have answered your questions in detail in our response, and feel free to let us know if you have any additional questions before the discussion period ends.
>
> Since the discussion period is closing very soon and none of the other reviewers have responded to us, we sincerely hope that you can positively support us during the final discussion with the AC. Thank you!
>
> Sincerely,
> Paper 9785 Authors

---

### Official Review · Reviewer_Ygz6 · 2022-07-12

**Rating:** 6
**Confidence:** 4
**Soundness:** 3 good
**Presentation:** 3 good
**Contribution:** 2 fair

**Summary:**

The paper considers the neural network verification problem where it builds on
bound propagation methods to account for cutting planes generated by MILP
solvers. Experimental results on a number of benchmarks from VNN-COMP 20201
show improved performance over the SoA.

**Questions:**

Please see comments on Weaknesses.

**Limitations:**

These are sufficiently addressed.

**Strengths And Weaknesses:**

Strengths.

- First work that I am aware of that uses cutting planes from MILP solvers in a
  bound propagation setting.

- It is good to see that this works in practice.

Weaknesses.

- Highly incremental to previous work. The technical novelty is the addition of
  cuts to a previously developed framework, which while is good to see that it
  works, it presents limited originality and overall contribution.

- There is no ablation study that compares GCP-CROWN with cuts and GCP-CROWN
  without cuts. The authors should make precise the contribution of the cuts to
  the overall performance (while there are comparisons with different variants
  of crown, an ablation study on GCP-CROWN would make the actual contribution
  more clear).

- Related work on MILP-based verification which strengthens the MILP
  formulation  with custom cuts is not discussed:  E. Botoeva et al., Efficient
  Verification of ReLU-Based Neural Networks via Dependency Analysis, AAAI
  2020.

---

> ### Author Response · Authors · 2022-08-02
> **Thank you for the valuable feedback and we have addressed the misunderstandings and questions below**
>
> We thank the reviewer for the constructive comments and valuable questions. We feel that most of the concerns are largely misunderstandings of our work, and we hope the reviewer can reevaluate our paper based on our response below:
>
> 1. We respectfully disagree that our work is highly incremental. Our work is extremely valuable to the neural network verification community because:
>
>    - Our work presents the **first efficient and GPU-accelerated approach to handle general cutting planes for neural network verification**. Effective and efficient cutting planes (which are very successful in integer programming solvers) have always been a missing part in NN verification and our work bridges this fundamental gap (line 42-64 in Intro). Specifically, the classical bound propagation framework is efficient but cannot handle general cutting planes. Our method enables the usage of general cutting planes in the GPU-accelerated bound propagation **for the first time**;
>    - Our general bound propagation method enables the future development of efficient general cutting planes for NN verification and **opens up entirely new opportunities to solve hard verification problems using GPU-accelerated cutting planes solvers**, not just limited to the MIP based cutting planes discussed in our paper;
>    - Experimentally, we significantly improved the results over SOTA verifiers on many benchmarks. Notably, we completely solved the oval20 benchmark where **none of existing tools can do** and also **doubled the number of verified instances** on oval21 benchmark compared to last year’s VNN-COMP winner. This is *not* an incremental improvement. We will make our contributions more clear in our revision.
>
> 2. Ablation study: Without cuts, our formulation and code base become the same as beta-CROWN.  “GCP-CROWN without cuts” (the ablation study mentioned by reviewer) **has already been reported** in our paper under the name beta-CROWN in Table 1, Figure 2, Table 2 and Table 3. Our implementation is a direct extension of beta-CROWN and when no cuts are added, it becomes vanilla beta-CROWN.  We clarified this point at the beginning of Section 4 in our revised version.
>
> 3. Thank you for pointing out this paper and we have added it to the related work section as well as in experiments. The paper (E. Botoeva et al.) is different in perspectives and contributions compared to ours. Their aim was to *find* a **specialized cutting plane** for NN verification and *solve* the verification problem with this cutting plane using a standard MILP solver. The focus of our paper is to *solve* the verification problem with **any types of cutting planes** using an **efficient and GPU accelerated bound propagation method**. In our paper, we use a MILP solver to *find* cuts as a showcase (not to *solve* the entire verification problem, which **should not be confused with the use of MILP solver in E. Botoeva et al.**), however our method is not limited to this specific method and can be combined with any other ways of generating cutting planes, including those proposed by (E. Botoeva et al.). So our contributions are orthogonal to theirs. We will add detailed discussions to the related work section in our final version.
>
> **Experimentally, we also added (E. Botoeva et al.) into comparison**. The original tool (venus) proposed in (E. Botoeva et al.) can handle fully connected networks only and cannot run any of our benchmarks (all our models involve convolutional layers). We thus run a stronger and most recent version of their tool (Venus 2) with convolutional layer support (added very recently, after NeurIPS submission deadline). We added results for VNN-COMP 2021 and SDP-FO benchmarks to Table 2, 3 and we consistently outperform their approach in all benchmarks.
>
> We’ve uploaded a revised version of our paper, but due to page limits we had to keep the additions short (highlighted in blue). If our paper is accepted, we will have one additional page to further extend the discussions requested by the reviewers.
>
> We hope the reviewer now can see why our paper is important for the field of neural network verification, and is clear about the ablation study (already reported in paper) and the relation to previous work (E. Botoeva et al.). Please kindly reevaluate our paper based on our response and let us know if you have any future questions. Thank you.

---

> ### Author Response · Authors · 2022-08-08
> **Thank you again for the helpful comments, and we hope to further discuss with you before the discussion period is closed**
>
> Dear Reviewer Ygz6,
>
> We thank you again for your helpful reviews. In our response, we have pointed out the misunderstanding on ablation study (which was already included), discussed E. Botoeva et al. as suggested, and also added new experimental comparisons to E. Botoeva et al.
>
> In addition, we respectfully disagree that our work is highly incremental - our work presents the **first efficient and GPU-accelerated** approach to handle **general cutting planes for neural network verification**, opens up entirely new opportunities to solve hard verification problems using GPU-accelerated cutting planes solvers. Experimentally it is also very successful: we completely solved the oval20 benchmark where **none of existing tools can do** and also **doubled the number of verified instances** on oval21 benchmark compared to VNN-COMP 2021 winner. This is not an incremental improvement.
>
> Since the discussion period is closing soon, we hope the reviewer can reconsider our paper based on our response. Please let us know if you have any additional questions for us. Thank you for your help!
>
> Best regards,
> Paper 9785 Authors

---

> ### Author Response · Authors · 2022-08-09
> **Thank you again for the helpful review, and we hope to hear from you before the discussion period ends**
>
> Dear Reviewer Ygz6,
>
> We greatly appreciate your helpful review! Since the discussion period is closing within a few hours, we sincerely hope the reviewer can reevaluate our paper based on our response on the misunderstandings on ablation study and novelty, and also the added discussions and new baselines. Please kindly let us know if you have any additional questions and we will try our best to respond during the last few hours. Thank you.
>
> Sincerely,
> Paper 9785 Authors

---

### Meta-Review · Area_Chair_sPrq · 2022-08-24

**Recommendation:** Accept
**Confidence:** Certain

**Metareview:**

All the reviewers found the work to have promise, but there was concern about the novelty of the work. That said, the experimental results showcased the power of the approach; the authors are advised to put their work in the context of prior work.

Overall, there was a consensus that the paper deserves to be published and hence the recommendation.

**Award:**

No

---

### Decision · Program_Chairs · 2022-09-14

Accept